# Reliability of an interneuron response depends on an integrated sensory state

**May Dobosiewicz[1], Qiang Liu[1], Cornelia I Bargmann[1,2]***

[1]Lulu and Anthony Wang Laboratory of Neural Circuits and Behavior, The Rockefeller University, New York, United States; [2]Chan Zuckerberg Initiative, Palo Alto, United States

**Abstract** The central nervous system transforms sensory information into representations that are salient to the animal. Here we define the logic of this transformation in a *Caenorhabditis elegans* integrating interneuron. AIA interneurons receive input from multiple chemosensory neurons that detect attractive odors. We show that reliable AIA responses require the coincidence of two sensory inputs: activation of AWA olfactory neurons that are activated by attractive odors, and inhibition of one or more chemosensory neurons that are inhibited by attractive odors. AWA activates AIA through an electrical synapse, while the disinhibitory pathway acts through glutamatergic chemical synapses. AIA interneurons have bistable electrophysiological properties consistent with their calcium dynamics, suggesting that AIA activation is a stereotyped response to an integrated stimulus. Our results indicate that AIA interneurons combine sensory information using AND-gate logic, requiring coordinated activity from multiple chemosensory neurons. We propose that AIA encodes positive valence based on an integrated sensory state.

## Introduction

Sensory environments are complex, and can include multiple signals within and across sensory modalities. Individual stimuli and the combinations in which they occur may differentially signal the presence of beneficial, harmful, or neutral conditions. Accordingly, integration across sensory inputs is an essential function of the nervous system, and the logic of integration is an area of active investigation.

The compact *Caenorhabditis elegans* nervous system, with 302 neurons and a complete neuronal wiring diagram, is well-suited for studying sensory representation and integration (*White et al., 1986*; *Cook et al., 2019*). *C. elegans* uses volatile and water-soluble cues to detect its food and other relevant stimuli (*Bargmann, 2006*). Its highly developed chemosensory system senses a large number of attractive and repulsive compounds using over a thousand different G-protein-coupled chemoreceptors, many of which are expressed by eleven pairs of chemosensory neurons in the amphid sensory organ (*Robertson and Thomas, 2006*). Specific subsets of chemosensory neurons have reproducible functions in chemotaxis and avoidance of chemical stimuli, spontaneous locomotion, physiology, development, and lifespan (*Bargmann, 2006*). AWA and AWC sensory neurons, for example, drive chemotaxis toward a variety of attractive odors, whereas AWB sensory neurons drive avoidance of certain repellents (*Bargmann et al., 1993*; *Troemel et al., 1997*). Notably, sensory neurons may be either activated or inhibited by odors. For example, AWA is activated by the attractive odor diacetyl (*Shinkai et al., 2011*; *Larsch et al., 2013*), whereas AWC is inhibited by another attractive odor, isoamyl alcohol (*Chalasani et al., 2007*).

At the first layer of neuronal integration, the sensory neurons form abundant chemical and electrical synapses onto a half-dozen pairs of interneurons (*White et al., 1986*). The interneurons regulate behaviors such as reversal frequency, speed, and head turning during thermotaxis, chemotaxis, and spontaneous locomotion (*Iino and Yoshida, 2009*; *Tsalik and Hobert, 2003*; *Gray et al., 2005*;

*For correspondence:
cori@rockefeller.edu

Competing interests: The authors declare that no competing interests exist.

*Hendricks et al., 2012*). Interneuron responses are variable and complex, and can incorporate feedback from network states and motor systems as well as sensory input (*Gordus et al., 2015*; *Li et al., 2014*; *Hendricks et al., 2012*; *Kaplan et al., 2018*). Synaptic mechanisms linking interneurons to locomotor states have been defined, but the computations that interneurons perform to integrate multiple sensory inputs are unknown.

The AIA interneuron pair receives chemical or electrical synapses from all eleven pairs of amphid chemosensory neurons, suggesting an integrative function (*White et al., 1986*) (*Supplementary file 1*). At a behavioral level, AIA is implicated in the suppression of reversal behavior upon odor addition (*Larsch et al., 2015*), integration of attractive and repulsive stimuli (*Shinkai et al., 2011*), olfactory desensitization at short and long timescales (*Chalasani et al., 2010*; *Cho et al., 2016*), and regulation of spontaneous reversals (*López-Cruz et al., 2019*). Functional calcium imaging has demonstrated that AIA can be activated by the attractive odors diacetyl and isoamyl alcohol (*Larsch et al., 2013*; *Chalasani et al., 2010*), which are primarily sensed by AWA and AWC neurons, respectively (*Bargmann et al., 1993*). In both cases, AIA activity rises in the presence of an attractive food-related odor.

Here, we use an optogenetic approach to isolate the connections between individual sensory neurons and AIA. We find that AIA uses AND-gate logic to integrate sensory information. AIA is reliably activated only by coordinated sensory information from multiple neurons, and this activation is mediated by both chemical and electrical synapses. A bistable current-voltage relationship provides a biophysical mechanism for the nonlinear AIA response. Our results suggest that AIA represents a positive valence that is integrated across sensory inputs.

## Results

### Optogenetic activation of AWA sensory neurons elicits unreliable AIA calcium responses

AWA sensory neurons expressing genetically-encoded calcium indicators such as GCaMP respond to the bacterial odorant diacetyl with fluorescence increases indicating depolarization (*Shinkai et al., 2011*; *Larsch et al., 2013*; *Zaslaver et al., 2015*; *Hale et al., 2016*; *Larsch et al., 2015*; *Liu et al., 2018*). AWA calcium responses are concentration-dependent, with stronger and more rapid responses to 115 nM diacetyl than to 11.5 nM diacetyl, and a rapid rise followed by desensitization within 10 s at 1.15 μM diacetyl (*Larsch et al., 2015*) (*Figure 1A and C*). Diacetyl also elicits calcium transients in the AIA interneurons, with desensitization at high diacetyl concentrations (*Larsch et al., 2015*) (*Figure 1B and D*). AIA calcium transients are diminished in animals with defects in AWA development or in the AWA diacetyl receptor ODR-10 (*Larsch et al., 2015*).

To examine AWA-to-AIA synaptic communication in detail, we directly depolarized AWA using the light-activated channel Chrimson expressed under an AWA-selective promoter and recorded GCaMP responses in either AWA or AIA (*Klapoetke et al., 2014*) (*Figure 1A–B*, *Figure 1—figure supplement 1A–F*). Light-activated calcium transients were elicited within one second in Chrimson-expressing AWA neurons and desensitized only slightly over ten seconds. This activation required pre-exposure to retinal and expression of the Chrimson transgene (*Figure 1—figure supplement 1B–C*), and was comparable in magnitude to the response of AWA at or above 115 nM diacetyl.

To our surprise, AIA responses to AWA::Chrimson stimulation were significantly smaller than AIA responses to any concentration of diacetyl (*Figure 1B*). To understand the discrepancy between optogenetic stimulation and odor, we examined the dynamics of individual calcium responses instead of averaged traces (*Figure 1D and E*, *Figure 1—figure supplement 2B*, and *Figure 1—figure supplement 3A*) and established robust thresholding parameters to separate responses from non-responses (see Materials and methods and *Figure 1—figure supplement 2*). The GCaMP fluorescence baseline in AIA was stable, with little spontaneous activity, and nearly all AIA neurons responded to the higher odor concentrations (115 nM and 1.15 μM diacetyl) with an average delay of ~1 s relative to AWA activation (*Figure 1C, D and F*, and *Figure 1—figure supplement 3B–C*). At the lowest tested diacetyl concentration (11.5 nM) AIA responded with a lower probability (*Figure 1D–F*), and a delay of ~2 s relative to AWA responses (*Figure 1F*, *Figure 1—figure supplement 3B*). Thus AIA responses to diacetyl are coupled to odor and dose-dependent. These AIA properties differ from those of AIB interneurons, which have spontaneous calcium fluctuations and

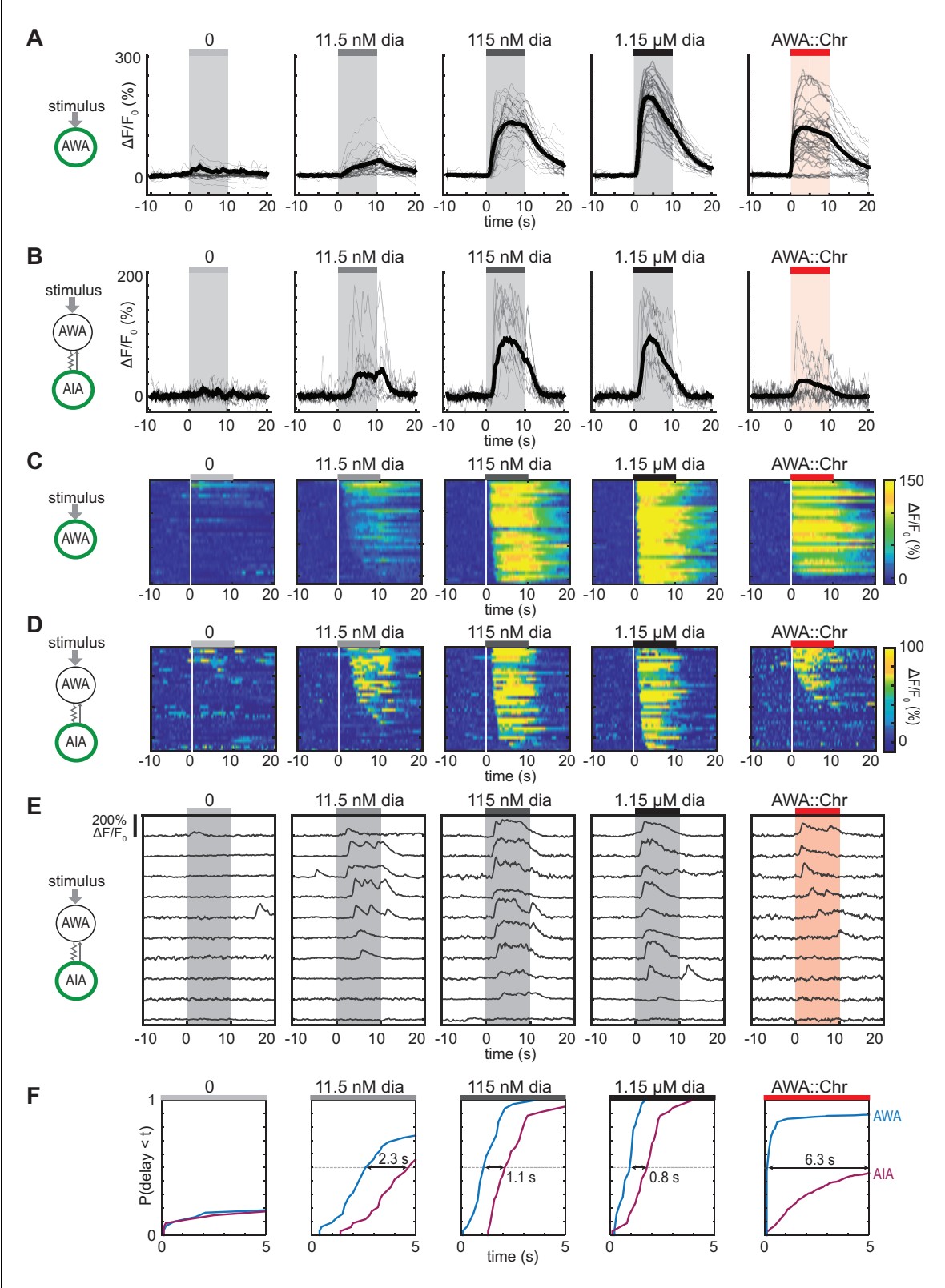

**Figure 1.** Optogenetic activation of AWA sensory neurons elicits unreliable AIA calcium responses. (**A and B**) AWA GCaMP2.2b (**A**) or AIA GCaMP5A (**B**) calcium responses to 10 s pulses of increasing concentrations of diacetyl and to AWA optogenetic stimulation. Bold lines indicate mean response, and light lines show individual traces. AWA traces to optogenetic stimulation were randomly downsampled to 40 traces from a complete set of 268 traces to match the number of odor traces and enhance visibility. AIA traces were randomly downsampled to 10 traces from a set of 34 (for 0–1.15 μM

*Figure 1 continued on next page*

Figure 1 continued

diacetyl) or 569 (for AWA::Chrimson stimulation). In all schematic diagrams, calcium was monitored in the neuron indicated in green, resistor symbols represent gap junctions, and thin arrows represent chemical synapses. (C and D) Heat maps of AWA (C) or AIA (D) calcium traces from (A) and (B), respectively. Responses to optogenetic stimulation were downsampled to 32 traces (C) or 34 traces (D) for visibility and to match sample sizes to diacetyl; see *Figure 1—figure supplement 3A* for complete data. Each heat map row represents a calcium trace to a single stimulus pulse; each animal received two stimulus pulses. Traces are ordered by response latency. (E) Representative AIA calcium traces to a given stimulus. Responses were sorted by response latency, binned into ten bins, then one trace was randomly selected from each bin for presentation. (F) Cumulative response time profiles of AWA and AIA responses representing response latencies and probability, without downsampling. Only first 5 s of stimulation are shown. Arrows indicate the delay between the time at which 50% of AWA neurons responded versus the time at which 50% of AIA neurons responded.

The online version of this article includes the following source data and figure supplement(s) for figure 1:

**Source data 1.** Source data for *Figure 1* and figure supplements.
**Figure supplement 1.** Experimental configuration and calibration of simultaneous GCaMP-Chrimson imaging conditions.
**Figure supplement 2.** Validation of AIA response thresholding procedure.
**Figure supplement 3.** Sequential imaging of AIA responses to odor or AWA::Chrimson stimulation.

variable sensory responses at all odor concentrations due to a strong effect of downstream motor state (*Gordus et al., 2015*; *Kato et al., 2015*).

At an individual trial level, AWA::Chrimson stimulation elicited AIA responses with a low probability and a delay, and these responses were much less robust than AIA responses elicited by diacetyl at a comparable level of AWA activation (*Figure 1D–F*). AWA::Chrimson elicited calcium increases in 85% of AWA sensory neurons within one second of optogenetic stimulation, resembling 115 nM diacetyl, but only 56% of the AIA interneurons were activated. Moreover, the AIA calcium transients were delayed by >6 s on average relative to the AWA response (*Figure 1F*, *Figure 1—figure supplement 3B*). Thus AIA calcium responses to AWA::Chrimson were unreliable.

Control experiments indicated that these differences were robust to transgenes or stimulus protocols. AWA::Chrimson animals had normal AIA and AWA responses to diacetyl, before or after light stimulation (*Figure 1—figure supplements 1D* and *3C*), and the AIA response latency was not correlated with GCaMP fluorescence levels or AWA::Chrimson transgene expression levels (*Figure 1—figure supplement 3D–E*). Animals were routinely subjected to two stimulus pulses of light or odor; these responses were slightly biased toward the first stimulus but largely independent (*Figure 1—figure supplement 3F–R*). Animals that responded to both stimulus pulses had correlated response magnitudes between the two pulses (*Figure 1—figure supplement 3J,N and R*). However, the latencies of responses were not correlated between the first and second pulse of 11.5 nM diacetyl or AWA::Chrimson stimulation, indicating that reliability is primarily a trial-to-trial property and not due to variation between animals (*Figure 1—figure supplement 3I,M and Q*).

Close examination of the AIA calcium signals revealed that the response rise dynamics to diacetyl or AWA::Chrimson stimulation were similar; the major differences were in their probability and latency. When aligned to the beginning of an AIA calcium response, positive AIA trials had similar rise times, whether elicited by optogenetic stimulation or by diacetyl (*Figure 1—figure supplement 2C–D*). These stereotyped properties were robust across experiments and thresholding parameters (*Figure 1—figure supplement 2*).

## Gap junctions mediate AWA-to-AIA communication

AWA cell fate mutants (*odr-7*) and AWA diacetyl receptor mutants (*odr-10*) have diminished AIA interneuron responses to diacetyl (*Larsch et al., 2015*; *Sengupta et al., 1994*; *Sengupta et al., 1996*). At the individual trial level, AIA interneuron responses to high (1.15 μM) diacetyl were less reliable in AWA-defective mutants than in wild type (*Figure 2A–B*). These results indicate that AWA is necessary for strong and reliable AIA interneuron responses to diacetyl, although the optogenetic experiments indicate that AWA activation is not sufficient for reliable AIA responses.

The *C. elegans* wiring diagram predicts the existence of gap junctions between AWA and AIA neurons, and a recent reanalysis additionally predicts a chemical synapse from AWA to AIA (*White et al., 1986*; *Cook et al., 2019*). To assess the importance of the potential chemical synapse, we inhibited AWA synaptic vesicle release by expressing the tetanus toxin light chain, which cleaves the synaptic vesicle protein synaptobrevin (*Schiavo et al., 1992*; *Macosko et al., 2009*). *AWA::TeTx* animals and wild type animals had indistinguishable AIA responses to diacetyl and AWA::Chrimson

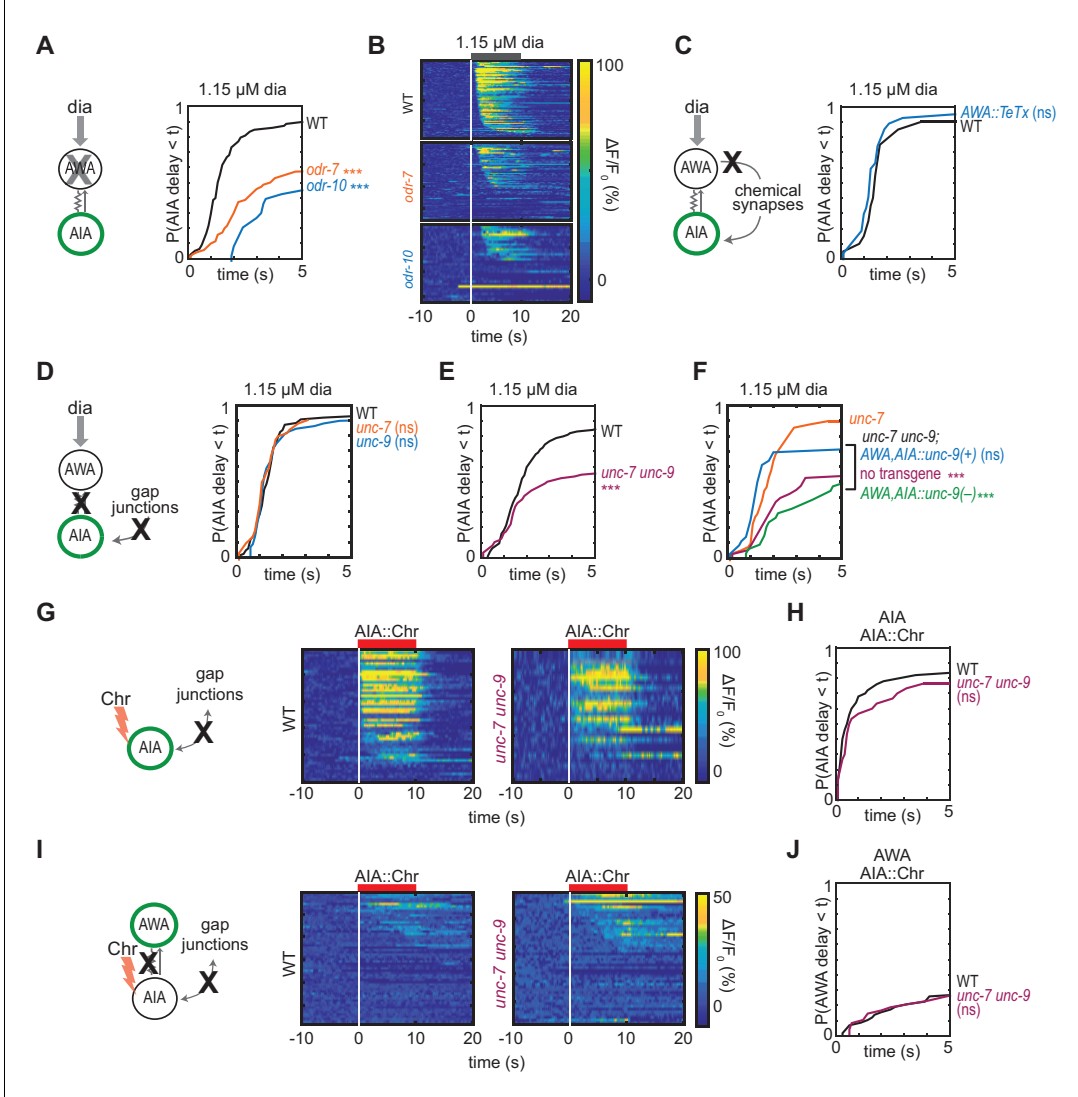

**Figure 2.** Gap junctions mediate AWA-to-AIA communication. (A, C, D and E) Cumulative response time profiles of AIA responses to 1.15 µM diacetyl in WT versus *odr-7* animals (AWA cell fate mutants) (A), *odr-10* animals (AWA diacetyl receptor mutants) (A), animals expressing a transgene encoding Tetanus Toxin Light Chain A (TeTx) in AWA (C), *unc-7* or *unc-9* animals (innexin mutants) (D), and *unc-7 unc-9* double mutants (E). (B) Heat maps of AIA responses to 1.15 µM diacetyl in WT, *odr-7*, and *odr-10* animals from (A). (F) *unc-9* innexin rescue in AWA and AIA. Cumulative response time profiles of AIA responses to 1.15 µM diacetyl in *unc-7* innexin mutants, *unc-7 unc-9* double mutants, *unc-7 unc-9; AWA,AIA::unc-9(+)* transgenic rescue animals, and *unc-7 unc-9; AWA,AIA::unc-9(fc16)* transgenic control animals. (G and I) AIA (G) and AWA (I) responses to 10 s pulses of AIA::Chrimson stimulation in WT and *unc-7 unc-9* animals; one row per calcium trace. Note that scale bar in (I) differs from scale bar in *Figure 1C*. (H) Cumulative response time profiles of AIA responses shown in (G). (J) Cumulative response time profiles of AWA responses shown in (I). Asterisks refer to Kolmogorov-Smirnov test significance versus WT (A, C, D, E, H and J) or versus *unc-7* (F) over full 10 s stimulus pulse. ns: not significant; ***: p<0.001. See *Supplementary file 2* for sample sizes and test details. Additional heat maps of data from *Figure 2* appear in *Figure 2—figure supplement 1*.

The online version of this article includes the following source data and figure supplement(s) for figure 2:

**Source data 1.** Source data for *Figure 2* and figure supplement.

**Figure supplement 1.** Gap junctions contribute to AWA-to-AIA communication, additional data.

stimulation, indicating that AWA does not require chemical synapses to activate AIA (*Figure 2C*, *Figure 2—figure supplement 1A*). Moreover, AWA::Chrimson stimulation and diacetyl strongly activated AIA in the synaptic transmission mutants *unc-13* and *unc-18*, providing further evidence that chemical synapses are not necessary for AWA to AIA signaling (*Figure 3*, discussed below).

We next asked whether AWA-to-AIA communication requires gap junctions, which are composed of innexin proteins in invertebrates. AWA and AIA express two innexin genes that contribute to

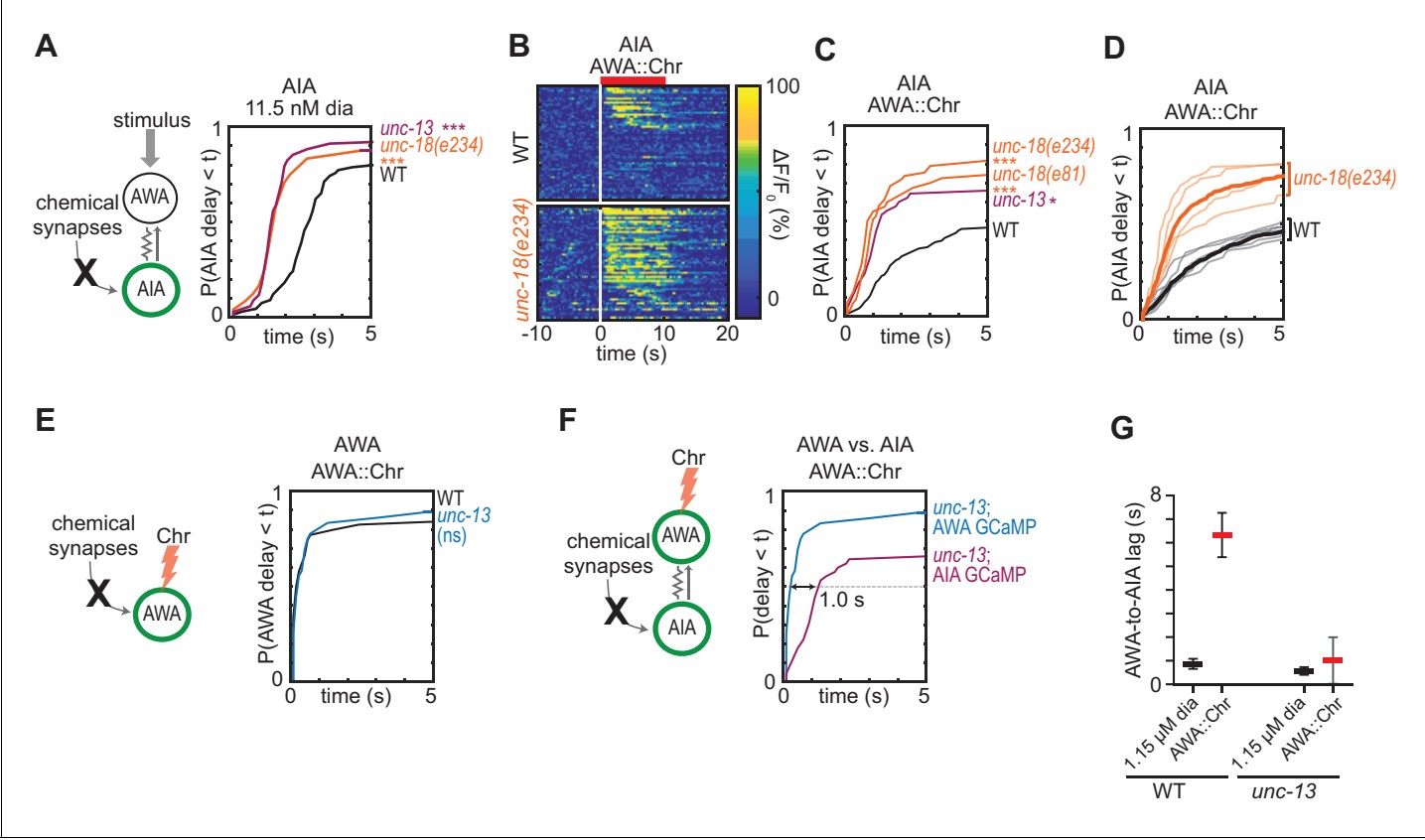

**Figure 3.** Chemical synapses inhibit AIA. (A) Cumulative response time profiles of AIA responses to 11.5 nM diacetyl in WT versus *unc-13(e51)* and *unc-18(e234)* animals (synaptic transmission mutants). (B) Heat maps of AIA responses to AWA::Chrimson stimulation in WT and *unc-18(e234)* animals. WT data were randomly downsampled to 57 traces for visibility and to match number of *unc-18(e234)* traces; data shown for single experiment block. See *Figure 3—figure supplement 1B* for pooled data from all experiments. (C) Cumulative response time profiles of AIA responses to AWA::Chrimson stimulation in WT versus *unc-13, unc-18(e234),* and *unc-18(e81)* animals. (D) Cumulative response time profiles of WT and *unc-18(e234)* response time profiles, combined over all experiments. Thick lines represent distributions of all data, faint lines represent distributions from individual experimental blocks. (E) Cumulative response time profiles of AWA responses to AWA::Chrimson stimulation in WT versus *unc-13* animals. (F) Cumulative response time profiles of AWA and AIA responses to AWA::Chrimson stimulation in *unc-13* animals. Arrow indicates the delay between the time at which 50% of AWA versus 50% of AIA neurons have responded. (G) Delay between the time at which 50% of AWA versus 50% of AIA neurons responded to 1.15 μM diacetyl and AWA::Chrimson stimulation in WT and *unc-13* animals. WT responses are the same as in *Figure 1—figure supplement 3B*. Bars are mean ± SEM. Asterisks refer to Kolmogorov-Smirnov test significance versus WT over full 10 s stimulus pulse. ns: not significant; *: p<0.05; ***: p<0.001. See *Supplementary file 2* for sample sizes and test details. Additional heat maps of data from *Figure 3* appear in *Figure 3—figure supplement 1*. The online version of this article includes the following source data and figure supplement(s) for figure 3:

**Source data 1.** Source data for *Figure 3* and figure supplement.
**Figure supplement 1.** Chemical synapses inhibit AIA, additional data.

many gap junctions, *unc-7* and *unc-9*, among other innexins (*Cao et al., 2017*; *Bhattacharya et al., 2019*). AIA responses to diacetyl were unaffected by single *unc-7* or *unc-9* mutants, but were less reliable in *unc-7 unc-9* innexin double mutants than in wild type (*Figure 2D–F* and *Figure 2—figure supplement 1B,C and E*), recapitulating the defects observed in AWA cell fate and receptor mutants (*odr-7, odr-10*; *Figure 2A*). AIA responses to AWA::Chrimson stimulation were also less reliable in *unc-7 unc-9* innexin double mutants (*Figure 2—figure supplement 1D*). Expressing an *unc-9* cDNA in AWA and AIA neurons of *unc-7 unc-9* double mutants rescued the AIA responses to diacetyl, but an *unc-9* cDNA with an inactivating point mutation did not rescue (*Figure 2F*, *Figure 2—figure supplement 1E*). Together, these results indicate that AWA signals to AIA via gap junctions.

Gap junctions can transmit information symmetrically or asymmetrically between neurons based on properties including voltage-dependent rectification, differential subunit expression between

cells, and differential phosphorylation (*Goodenough and Paul, 2009*). To test whether the predicted electrical synapse between AWA and AIA is bidirectional, we optogenetically stimulated AIA interneurons with a Chrimson transgene and recorded the resulting responses in AWA sensory neurons. AIA responded rapidly and robustly to direct optogenetic stimulation (*Figure 2G and H*), but AWA responded infrequently, with only small-magnitude calcium responses, to AIA stimulation (*Figure 2I and J*). The AWA response to AIA stimulation was unchanged in *unc-7 unc-9* innexin double mutants, and slightly increased in synaptic transmission mutants, suggesting that multiple synapse types contribute to weak AIA-to-AWA communication (*Figure 2G–2J*, *Figure 2—figure supplement 1F–I*). Together, these results suggest that AWA-to-AIA gap junctions preferentially mediate antero-grade information flow from sensory neurons to interneurons.

## Chemical synapses inhibit AIA

The rapid AIA response to direct AIA optogenetic stimulation indicates that its delayed response to sensory stimuli is not caused by slow intrinsic calcium dynamics, but by other circuit elements. Diacetyl can activate AIA, albeit less reliably, in mutants that lack AWA function or *unc-7* and *unc-9* innexins. The residual AIA diacetyl response predicts that additional diacetyl-sensing neurons communicate with AIA, most likely through chemical synapses. Indeed, AIA receives chemical synapses from many chemosensory neurons (*Supplementary file 1*).

We asked how chemical synapses impact AIA responses by examining mutants in *unc-13* and *unc-18*, both of which are deficient in synaptic vesicle release (*Richmond, 2007*). Unexpectedly, AIA responses to diacetyl were faster and more reliable in animals with defective chemical synapses (*Figure 3A*, *Figure 3—figure supplement 1A*). This effect was most evident at low diacetyl concentrations, where the synaptic mutants substantially decreased the latency of the AIA response (*Figure 3A*). A trend toward faster AIA responses at the higher diacetyl concentrations only reached significance for one of three tested mutants, perhaps reflecting a ceiling effect (*Figure 3—figure supplement 1A*). Inactivation of dense core vesicle release with an *unc-31* mutation did not alter AIA reliability, demonstrating specificity of the effect (*Richmond, 2007*) (*Figure 3—figure supplement 1E*). Chemical synapses are thus net inhibitory onto AIA.

The increased reliability of AIA responses in *unc-13* and *unc-18* mutants was even more striking when AWA was stimulated with AWA::Chrimson (*Figure 3B–3D*, *Figure 3—figure supplement 1B*). In the synaptic mutants, half of the AIA neurons responded to AWA::Chrimson within 1.2 s, a dramatic decrease in latency compared to wild type (>6 s) (*Figure 3C*). The AWA-to-AIA delay in synaptic mutants after AWA::Chrimson stimulation resembled the delay to high concentrations of diacetyl (*Figure 3F–G*). Control experiments demonstrated that direct AWA responses to optogenetic stimulation did not increase in *unc-13* synaptic mutants (*Figure 3E*, *Figure 3—figure supplement 1C–D*). Chemical synapses thus inhibit AIA interneurons primarily by decreasing the reliability of AIA's response at a given level of AWA activation.

## Glutamatergic sensory neurons cooperate to inhibit AIA

Eighteen pairs of neurons form chemical synapses onto AIA, six of which are sensory neurons that use glutamate as a neurotransmitter (*White et al., 1986*; *Cook et al., 2019*; *Serrano-Saiz et al., 2013*). Glutamate hyperpolarizes AIA, likely by activating glutamate-gated chloride channels, so these glutamatergic sensory neurons are plausible sources of the synaptic inhibition of AIA (*Chalasani et al., 2010*; *Shinkai et al., 2011*; *López-Cruz et al., 2019*). We selectively inhibited glutamate release using a CRISPR-edited version of the endogenous *eat-4* locus, which encodes the major vesicular glutamate transporter in *C. elegans*; the edited gene enables cell-specific *eat-4* excision with flippase recombinase (*Lee et al., 1999*; *López-Cruz et al., 2019*) (*Figure 4A and B*). Selective excision of *eat-4* in four *tax-4*-expressing sensory neurons, AWC, ASE, ASK, and ASG, allowed AWA::Chrimson stimulation to evoke reliable AIA responses similar to those in *unc-18* synaptic transmission mutants (*Figure 4C*, *Figure 4—figure supplement 1A*). This effect was not observed with either the modified *eat-4* locus or flippase expression alone (*Figure 4—figure supplement 1A–B*).

No individual sensory pair accounted for the full effect of preventing glutamate release from AWC, ASE, ASK, and ASG together (*Figure 4D–4H*, *Figure 4—figure supplement 1A and C*), although a significant partial effect was observed upon inhibition of glutamate release from ASK alone (*Figure 4D*). As previous studies implicated AWC and ASE in diacetyl responses (*Larsch et al.,*

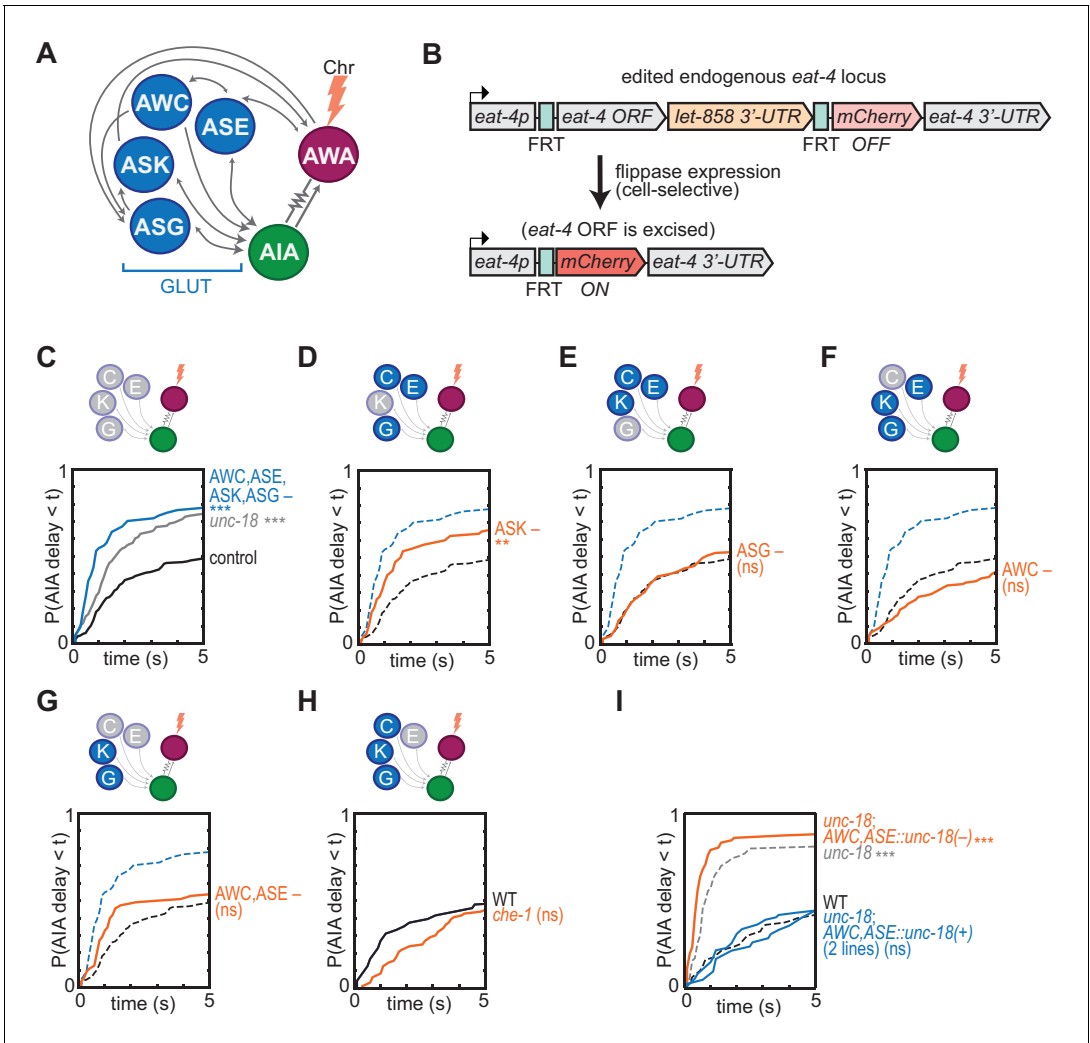

**Figure 4.** Glutamatergic sensory neurons cooperate to inhibit AIA. (A) Simplified diagram of connections between AWA, AIA and four glutamatergic sensory neurons, based on *White et al. (1986)*. (B) Schematic of cell-selective glutamate knockout genetic strategy (*López-Cruz et al., 2019*). The *eat-4* locus is excised only in the presence of flippase. ORF: open reading frame; UTR: untranslated region; FRT: flippase recombinase target. (C–H) Cumulative response time profiles of AIA responses to AWA::Chrimson stimulation in various animals lacking either glutamate release or cellular function of specific sensory neurons. For (D–G), dotted black and blue lines are control and *eat-4-FRT; AWC,ASE,ASK,ASG::nFlippase*, respectively, from (C). (C) Control (*eat-4-FRT* genetic background with no flippase expression), *unc-18*, and *eat-4-FRT; AWC,ASE,ASK,ASG::nFlippase* animals. (D) *eat-4-FRT; ASK::nFlippase* animals. (E) *eat-4-FRT; ASG::nFlippase* animals. (F) *eat-4-FRT; AWC::nFlippase* animals. (G) *eat-4-FRT; AWC+ASE::nFlippase* animals. (H) WT and *che-1* animals (ASE cell fate mutants). (I) Cumulative response time profiles of AIA responses to AWA::Chrimson stimulation in WT, *unc-18* animals, *unc-18; AWC,ASE::unc-18(+)* transgenic rescue animals (two lines), and *unc-18; AWC,ASE::unc-18(e234)* transgenic control animals. Asterisks refer to Kolmogorov-Smirnov significance versus *eat-4-FRT* controls (C–G) or WT (H, I) over full 10 s stimulus pulse. ns: not significant; \*\*: p<0.01; \*\*\*: p<0.001. See *Supplementary file 2* for sample sizes and test details. Heat maps of data from *Figure 4* appear in *Figure 4—figure supplement 1*. Additional representations of data from *Figures 1–4* appear in *Figure 4—figure supplement 2*.

The online version of this article includes the following source data and figure supplement(s) for figure 4:

**Source data 1.** Source data for *Figure 4* and figure supplements.

**Figure supplement 1.** Controls and heat maps for FRT-FLP recombination.

**Figure supplement 2.** Additional representations of AIA data from *Figures 1–4*.

*2015*), we further examined the effect of chemical synapses from these two neurons. As noted above, the *unc-18* mutant has increased reliability of AIA responses to AWA::Chrimson stimulation, but in two independent lines in which *unc-18* was selectively rescued in ASE and AWC, AIA interneuron responses were restored to unreliable responses resembling wild type (*Figure 4I, Figure 4—figure supplement 1D*). This effect was not observed with a control transgene that encoded the

inactive *unc-18(e234)* mutant in AWC and ASE. Synaptic vesicle release from AWC and ASE is therefore sufficient to inhibit AIA activation by AWA. In summary, multiple sensory neurons, including ASK and at least one of AWC and ASE, can release glutamate to inhibit AIA activation.

## Multiple sensory neurons detect diacetyl

To explain these results, we hypothesized that glutamatergic sensory neurons tonically inhibit AIA, and are inhibited when diacetyl is present to disinhibit AIA. This hypothesis agrees with previous calcium imaging studies showing that ASK and AWC are active at rest and inhibited by amino acids, pheromones, and certain odors (*Wakabayashi et al., 2009*; *Macosko et al., 2009*; *Zaslaver et al., 2015*; *Chalasani et al., 2007*; *Kato et al., 2014*). To extend this observation to diacetyl, we expressed the genetically-encoded calcium indicator GCaMP5A in ASK and AWC, and GCaMP3 in ASE, and determined that ASK and AWC were inhibited by the addition of 1.15 µM diacetyl, whereas ASE was activated by diacetyl at the same concentration (*Figure 5A–5C*, *Figure 5—figure supplement 1A–C*). All responses were dose-dependent, with responses that were weaker (ASK) or absent (AWC, ASE) at 11.5 nM diacetyl (*Figure 5A–5C*, *Figure 5—figure supplement 1A–C,E–G*). ASH, another glutamatergic sensory neuron that forms chemical synapses onto AIA, did not respond to 1.15 µM diacetyl (*Figure 5—figure supplement 1D and H*), demonstrating neuronal specificity of the response.

*C. elegans* sensory neurons are abundantly interconnected by chemical synapses (*White et al., 1986*). Diacetyl responses in ASK and AWC were unchanged in *unc-18* synaptic transmission mutants, suggesting that these neurons detect diacetyl directly (*Figure 5—figure supplement 1I–J, L–M*). By contrast, diacetyl responses in ASE were eliminated in *unc-18* mutants, suggesting that ASE senses diacetyl indirectly via another sensory neuron (*Figure 5D*, *Figure 5—figure supplement 1K*). AWA was not the source of the diacetyl response in ASE, as the response was preserved in an *odr-10* diacetyl receptor mutant (*Figure 5D*, *Figure 5—figure supplement 1K*). ASE receives synapses from many other sensory neurons, but as ASE was not essential or sufficient for AIA activation (*Figure 4G–H*), we did not pursue it further. Neither ASK, AWC, nor ASE showed a calcium response upon AWA::Chrimson stimulation (*Figure 5—figure supplement 1N–O*).

In summary, AIA responses are unreliable upon AWA::Chrimson stimulation, which activates only AWA; more reliable to 11.5 nM diacetyl, which activates AWA and inhibits ASK; and highly reliable to 1.15 µM diacetyl, which activates AWA and ASE and inhibits AWC and ASK, leading to disinhibition of AIA (*Figure 5H*). To ask whether the reduced reliability of AIA responses to 11.5 nM compared to 1.15 µM diacetyl was associated with reduced reliability of sensory neuron responses at low concentrations, we simultaneous recorded AIA, AWA and ASK calcium activity in a small number of animals while delivering pulses of 11.5 nM or 1.15 µM diacetyl. AIA was activated by 12/12 pulses of 1.15 µM diacetyl, and by 6/8 pulses of 11.5 nM diacetyl (*Figure 5—figure supplement 2A–B*). In each successful trial, AWA was activated, ASK was inhibited, and AIA was activated. In the two trials in which 11.5 nM diacetyl failed to elicit AIA responses, it elicited an AWA response but failed to elicit an ASK response (*Figure 5—figure supplement 2B*), consistent with a role for multiple sensory inputs in AIA activation.

## Combinatorial activation of AIA by isoamyl alcohol-sensing neurons

Since the reliability of the AIA response required inputs from multiple sensory neurons, we asked whether the same logic applied for other odors. Based on previous ablation studies, chemotaxis to diacetyl requires AWA at low concentrations, with a redundant role for AWC at high concentrations (*Chou et al., 2001*), whereas chemotaxis to another odor and bacterial metabolite, isoamyl alcohol, requires AWC with a minor contribution from AWA (*Bargmann et al., 1993*; *Worthy et al., 2018*). Both AWC and AWA respond to isoamyl alcohol with calcium transients (*Larsch et al., 2013*), suggesting that study of this second odor could test the generality of the AIA activation model.

Previous work showed that AWC is inhibited by 9 and 90 µM isoamyl alcohol (*Larsch et al., 2013*; *Yoshida et al., 2012*; *Gordus et al., 2015*). We found that AWC was also inhibited by 0.9 µM isoamyl alcohol, and that AWA was activated by isoamyl alcohol in a graded fashion at 0.9, 9, and 90 µM isoamyl alcohol (*Figure 5F–G*, *Figure 5—figure supplement 3A–B,D–E*). ASK was not inhibited by isoamyl alcohol at any tested concentration (*Figure 5E*, *Figure 5—figure supplement 3C and F*).

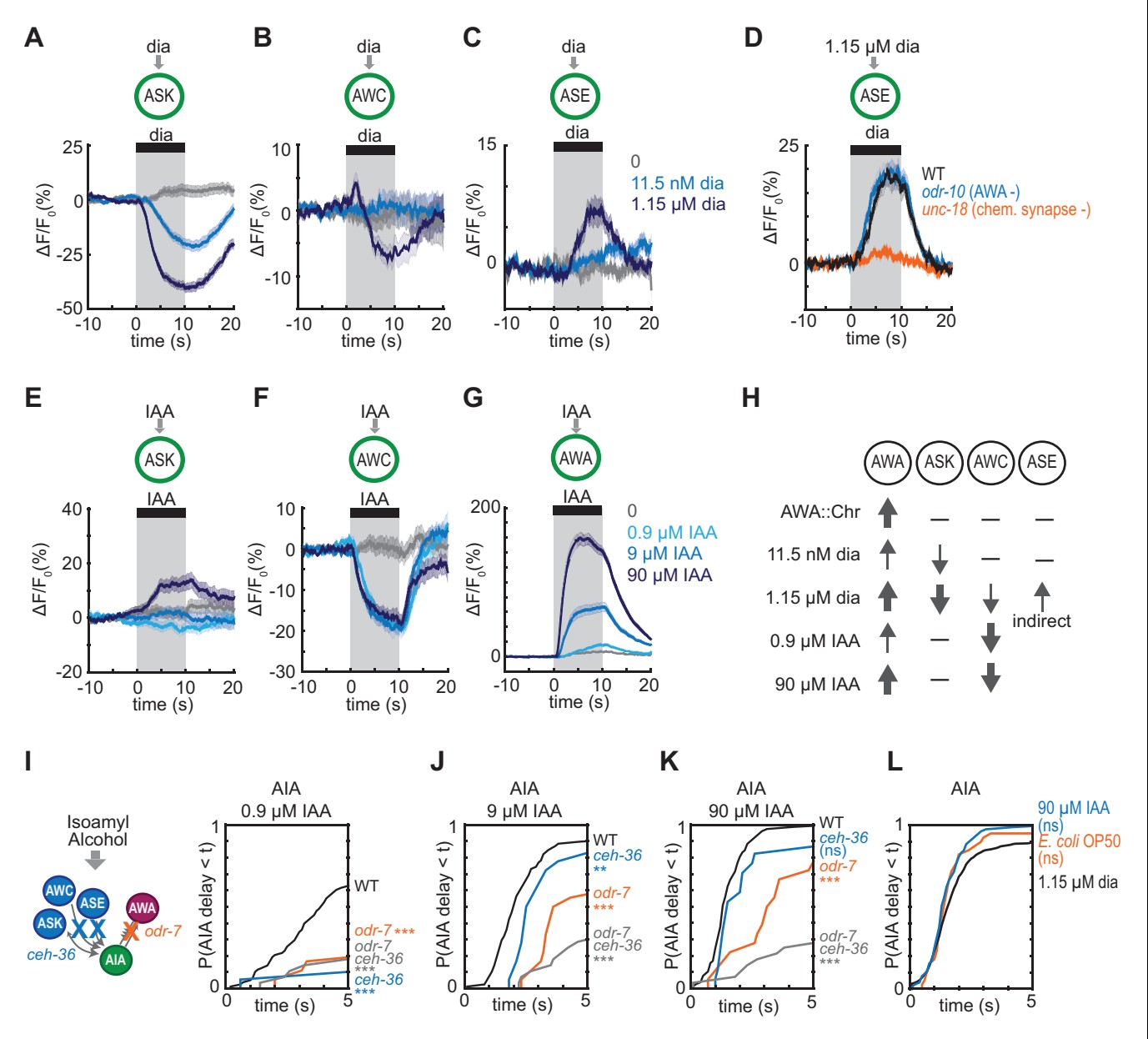

**Figure 5.** Multiple sensory neurons detect diacetyl and isoamyl alcohol. (**A – C**) Mean ASK (**A**), AWC (**B**), and ASE (**C**) responses to 10 s pulses of buffer (0) or 11.5 nM or 1.15 µM diacetyl. ASK: n = 82–115; AWC: n = 52–60; ASE: n = 42–54. Shading indicates ± SEM. (**D**) Mean ASE responses to 1.15 µM diacetyl in WT versus *unc-18* animals (synaptic transmission mutants) and *odr-10* animals (AWA diacetyl receptor mutants). ASE responses in *unc-18* animals are greatly diminished. Shading indicates ± SEM. (**E – G**) Mean ASK (**E**), AWC (**F**), and AWA (**G**) responses to 10 s pulses of buffer (0), 0.9 µM, 9 µM, or 90 µM isoamyl alcohol. ASK: n = 60; AWC: n = 42; AWA: n = 78–80. Shading indicates ± SEM. (**H**) Summary of sensory neuron responses to various stimuli. Upward arrows indicate activation; downward arrows indicate inhibition. Arrow thickness reflects response magnitude. (**I – K**) Cumulative response time profiles of AIA responses to 0.9 µM (**I**), 9 µM (**J**), or 90 µM (**K**) isoamyl alcohol in WT versus *odr-7* animals (AWA cell fate mutants), *ceh-36* animals (AWC and ASE cell fate mutants), and *odr-7 ceh-36* animals. (**L**) Cumulative response time profiles of AIA responses to 1.15 µM diacetyl, 90 µM isoamyl alcohol, and *E. coli* OP50 bacteria-conditioned medium. Asterisks refer to Kolmogorov-Smirnov test significance versus buffer over full 10 s stimulus pulse. ns: not significant; *: p<0.05; **: p<0.01; ***: p<0.001. See **Supplementary file 2** for sample sizes and test details.

The online version of this article includes the following source data and figure supplement(s) for figure 5:

**Source data 1.** Source data for *Figure 5* and *Figure 5—figure supplements 1–3*.
**Figure supplement 1.** Controls for diacetyl activation of sensory neurons.
**Figure supplement 2.** Simultaneous recording of multiple neurons.
**Figure supplement 2—source data 1.** Source data for *Figure 5—figure supplement 2*.
**Figure supplement 3.** Controls and heat maps for isoamyl alcohol stimulation.

We then examined AIA interneuron responses to 0.9, 9, and 90 µM isoamyl alcohol. As with diacetyl, AIA calcium responses were unreliable at the lowest concentrations, but became more reliable, with a higher probability and a shorter latency, as isoamyl alcohol concentration increased (*Figure 5I–K*, *Figure 5—figure supplement 3K*). At all concentrations, AWC responded first and AIA responded later, near the time of AWA activation (*Figure 5—figure supplement 3G–I*). Interestingly, a different pattern held for diacetyl, where AWA responded first and AIA responded later, near the time of ASK inhibition (*Figure 5—figure supplements 2A–D* and *3J*).

To test the contributions of different sensory neurons to AIA activation by isoamyl alcohol, we monitored AIA responses in wild type, AWA cell fate mutants (*odr-7*), AWC and ASE cell fate mutants (*ceh-36*) (*Lanjuin et al., 2003*), and *odr-7 ceh-36* double mutants. *odr-7* mutants had unreliable AIA responses to isoamyl alcohol at all concentrations (*Figure 5I–5K*, *Figure 5—figure supplement 3K*). At the higher concentrations of 9 µM and 90 µM isoamyl alcohol, AIA responses were more reliable in the *odr-7* and *ceh-36* single mutants than in the *odr-7 ceh-36* double mutants, indicating that AWA and AWC are partly redundant for AIA activation. At the lower concentration of 0.9 µM isoamyl alcohol, all mutants had unreliable AIA responses, indicating that both AWA and AWC are independently required for AIA activation.

Both diacetyl and isoamyl alcohol represent bacterial food sources (*Choi et al., 2016*; *Worthy et al., 2018*). Previous work has shown that the more complex food stimulus of *Escherichia coli* OP50-conditioned medium activates AWA and ASE and inhibits AWC, ASK, and several other sensory neurons (*Zaslaver et al., 2015*). We found that OP50-conditioned medium elicited reliable AIA calcium responses with similar latency and rise dynamics to those elicited by high concentrations of diacetyl or isoamyl alcohol (*Figure 5L*, *Figure 5—figure supplement 3L–M*). All stimuli that evoked reliable AIA responses engaged multiple sensory neurons, including inhibition of at least one glutamatergic sensory neuron and the activation of AWA (*Figure 6D*).

## AIA has a nonlinear current-voltage relationship

The finding that AIA calcium responses require multiple sensory inputs suggests that AIA is performing a non-additive, nonlinear thresholding calculation. To seek biophysical evidence for this nonlinearity, we recorded AIA voltage responses to electrophysiological current injections in dissected animals using step and ramp injection protocols under current clamp. We found that AIA membrane potential was bistable, with one stable state near the resting potential of −80 mV and a second state near −20 mV (*Figure 6A*). A rapid transition between these two states occurred with a low current injection threshold near 2–3 pA, at which point the AIA membrane potential rose rapidly to −20 mV (*Figure 6A–C*). Subsequent increases in stimulus intensity did not greatly increase AIA membrane potential beyond −20 mV. Voltage-clamp recordings indicate that AIA bistability is associated with a region of high membrane resistance between −80 and −20 mV (*Figure 6—figure supplement 1A–B*).

Simultaneous recordings of membrane potential and GCaMP fluorescence in AIA during current injections revealed GCaMP fluorescence levels that were also bistable and stereotyped, with increases that were correlated with the voltage transition from −80 to −20 mV (*Figure 6C*). Thus AIA performs a nonlinear rather than an additive computation at a biophysical level.

## Discussion

### AIA uses AND-gate logic to integrate sensory information

Our work defines the contributions of individual sensory neurons to the activation of a downstream integrating neuron. Multiple *C. elegans* sensory neurons respond to an attractive odor such as diacetyl or isoamyl alcohol with cell type-specific dynamics and signs. The sensory neurons signal the presence of odor to AIA using electrical synapses (AWA) or glutamatergic chemical synapses (AWC, ASK). AIA does not respond reliably to input from a single sensory neuron. Rather, AIA is activated by coordinated inputs from multiple sensory neurons, functioning as an AND-gate that requires both activation by AWA, and disinhibition from glutamatergic neurons including AWC and ASK.

We propose the following model for AIA integration (*Figure 6D*). In the absence of food odor, glutamatergic sensory neurons release glutamate that activates glutamate-gated chloride channels on AIA (*López-Cruz et al., 2019*). The chloride current decreases AIA membrane resistance,

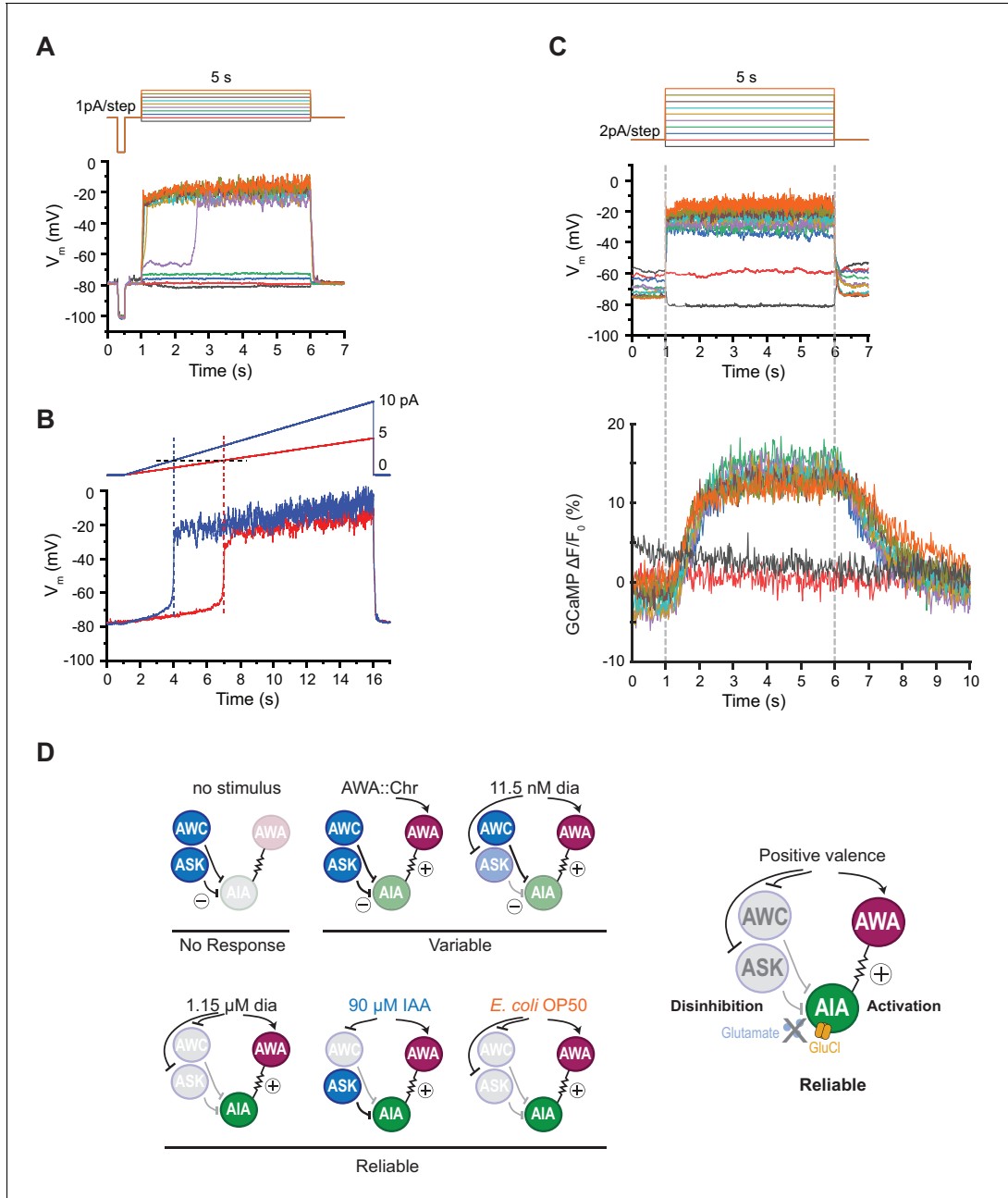

**Figure 6.** AIA neurons are bistable and act as a nonlinear AND-gate. (**A**) Representative example of membrane potential dynamics induced by current injection steps in current-clamped AIA neurons (n=45 AIA neurons recorded, all showing bimodal dynamics). Top, current injection protocol: a series of 5 s square pulses starting at −1 pA and increasing to 8 pA in 1 pA increments. Current injections between −1 and 2 pA had little effect on membrane potential ($V_m$), while the 3 pA step and all larger steps thereafter depolarized $V_m$ from the resting $V_m$ around −80 mV to a stable state of higher voltage around −20 mV. (**B**) Representative membrane potential dynamics (2 AIA neurons recorded) induced by current injection ramps at different slopes in AIA. Top, current injection protocol: two 15 s long ramping current injections from 0 pA to 5 pA (red) or 10 pA (blue) before returning to 0 pA, recorded from the same cell. AIA $V_m$ is abruptly depolarized from around −80 mV to −20 mV when the current injection ramps reaches the same threshold around 2 pA. (**C**) Representative example of simultaneous membrane potential recording (upper traces) and calcium imaging in the AIA neurite (lower traces). Top, current injection protocol: a series of 5 s square pulses starting at −2 pA and increasing to 16 pA in 2 pA increments. Both membrane potential and GCaMP signals show bimodal all-or-none dynamics. (**D**) Summary of sensory neuron and AIA responses to various stimuli, and a model for how AIA uses AND-gate logic to integrate sensory information. In the absence of food odor, glutamatergic sensory neurons release glutamate and activate glutamate-gated chloride channels on AIA, preventing AWA from activating AIA. In the presence of food odor, AWA is activated, glutamatergic sensory neurons are inhibited, and AIA is disinhibited and more sensitive to AWA depolarization, resulting in reliable activation. Weak odor or optogenetic stimuli engage a subset of sensory neurons, resulting in a variable and delayed response in AIA. With respect to

*Figure 6 continued on next page*

*Figure 6 continued*

the delay, full conditions for the AIA AND-gate may be met when weak stimuli coincide with spontaneous sensory neuron activity (*López-Cruz et al., 2019*; *Figure 5—figure supplement 2*).

The online version of this article includes the following source data and figure supplement(s) for figure 6:

**Source data 1.** Source data for *Figure 6A–B* and figure supplement.
**Source data 2.** Source data for *Figure 6C*.
**Figure supplement 1.** Current-voltage relationship in AIA.

preventing AIA activation via the AWA-AIA electrical synapse. In the presence of food odor, glutamatergic sensory neurons are inhibited, the glutamate-gated chloride channels close, and AIA disinhibition and increased AIA resistance make it more sensitive to depolarizing inputs. At the same time, food odors activate AWA, driving AWA depolarization and calcium action potentials (*Liu et al., 2018*), and transmission of this information through AWA-to-AIA gap junctions depolarizes AIA to the plateau potential of −20 mV.

The properties of the AND-gate and the contributing neurons explain both the low probability of an AIA response and the delayed response to weak odor stimuli. For both diacetyl and isoamyl alcohol, AIA activation lags behind the first sensory neuron to detect odor, matching the arrival of the second input. The sensory neurons, in turn, can contribute to this delay. At low odor levels, the inhibition of the glutamatergic sensory neurons reported by calcium imaging is graded and gradual. Similarly, the first AWA action potential lags behind stimulus onset by about a second (*Liu et al., 2018*). As a result, the synchronization of the disinhibitory and excitatory sensory inputs required for the AND-gate are delayed at low stimulus levels.

AIA membrane potential is bistable, with a sharp threshold that separates stable states near −80 mV and −20 mV. The depolarized state resembles the plateau potentials first reported in *C. elegans* RMD motor neurons (*Mellem et al., 2008*) and subsequently observed in a number of other *C. elegans* neurons that have a high resistance regime between stable low and high voltage states (*Liu et al., 2018*). The intrinsic bistability of AIA creates a threshold nonlinearity that shapes the response to synaptic inputs. The synaptic inputs are predicted to increase depolarizing current (through AWA gap junctions) and to increase membrane resistance (by closing glutamate-gated chloride channels), and therefore could have the multiplicative effect on voltage classically defined by Ohm's law. A preferential effect of AWA gap junctions on depolarizing current, and inhibitory chemical synapses on membrane resistance and shunting, provide a plausible mechanism for the AIA AND-gate.

The AND-gate is an established motif in transcriptional regulation (*Buchler et al., 2003*), and bears similarity to the related concept of coincidence detection in neural circuits (*Koch, 1999*). In contrast with computational models of neurons based on additive inputs onto the target neuron (*Dayan and Abbott, 2001*; *McCulloch and Pitts, 1943*), the AND-gate computation is nonlinear and multiplicative (*Koch, 1999*). In the context of AIA, this logical computation requires multiple sensory neurons to report the presence of an attractant, an integrative step that may filter out environmental noise. Based on our results with diacetyl and isoamyl alcohol, it appears that different combinations of sensory neurons can generate the disinhibition and excitation necessary for reliable AIA activation.

It is interesting to compare AIA to another well-characterized olfactory interneuron, AIB. AIB activity rises when odors such as isoamyl alcohol are removed, triggering aversive behaviors such as reversals and turns (*Chalasani et al., 2007*). The coupling of strong odor stimuli to AIB activity is less reliable than coupling to AIA: stimuli that drive AIA responses in over 90% of trials, drive AIB responses in only 57% of trials (*Gordus et al., 2015*). At a circuit level, AIB variability results from its integration of sensory input with the downstream motor state (*Gordus et al., 2015*), and AIB activity is highly correlated with the activity of other neurons that drive reversals (*Gordus et al., 2015*; *Kato et al., 2015*). AIA activity has not been studied systematically with respect to network states, although it appears to rise during transitions to forward locomotion (*Laurent et al., 2015*). In the *C. elegans* wiring diagram, AIA receives three times as many synapses from sensory neurons as from interneurons, and AIB receives equal numbers of synapses from sensory neurons and interneurons,

perhaps explaining why AIA activity is more closely coupled to sensory state, and less to motor state, than AIB (*White et al., 1986*). That said, motor states were not examined in this work, and they could contribute to the remaining variability in AIA activation. For example, even when the response probability to odors is high, the magnitude of the AIA calcium response can vary somewhat between trials, and neither AIA response magnitude nor response duration were analyzed systematically here. Further investigation could uncover input from network states on these and other quantitative aspects of the AIA response.

## Excitation and disinhibition by electrical and chemical synapses

In *C. elegans,* behavioral functions of many neurons are known – for example, AIA supports forward movement in the presence of odor, reversals upon odor removal, and aversive olfactory learning – but the associated synaptic computations are not as well understood. As predicted by the wiring diagram, our results indicate that AWA primarily excites AIA via an electrical synapse, in the context of chemical synapses from other neurons. Mixed chemical-electrical synapses are present in many escape circuits, including insect giant fibers, goldfish Mauthner cells, and the *C. elegans* backward command neurons (*Allen and Murphey, 2007*; *Phelan et al., 2008*; *Liu et al., 2017*). In these escape circuits, the same cells form chemical and electrical synapses with each other to maximize speed and efficiency. By contrast, AIA receives chemical and electrical synapses from different sensory neurons to enable integration.

At a synaptic level, the combination of excitatory and disinhibitory inputs of AIA resemble interactions at the glomeruli of the rat inferior olivary nucleus (*Pereda et al., 2013*). Principal cells in the inferior olive are coupled via electrical synapses, and are uncoupled when neurons from the deep cerebellar nuclei release GABA at adjacent chemical synapses. The inhibitory synapses shunt excitatory current flow at the electrical synapse (*Pereda et al., 2013*). Synchronized firing in this system has features of an AND-gate, with distinct states that depend on both electrical and chemical synapses.

At a circuit level, the roles of excitation and inhibition in an AND-gate differ from their roles in many circuits. Excitation and inhibition are most frequently observed as balanced inputs, tightly correlated in time and space, poising circuits to detect small changes with precision (*Okun and Lampl, 2008*; *Denève and Machens, 2016*). In the AND-gate logic employed by AIA, excitation and inhibition alternate, switching the neuron's activity state. A related disinhibition logic, with a more complex mechanism, is used in mammalian cortical and hippocampal circuits, where VIP-expressing interneurons disinhibit information flow through excitatory circuits by GABAergic inhibition of local inhibitory interneurons (*Pi et al., 2013*; *Turi et al., 2019*).

## AIA activity as a readout of positive valence

We suggest that AIA signals an integrated, positive sensory valence, as represented through the activity state of multiple sensory neurons. In addition to the odors studied here, further evidence for combinatorial integration by AIA comes from behavioral studies in which animals cross an aversive copper barrier, sensed by the glutamatergic ASH sensory neurons, to reach an attractive diacetyl source (*Shinkai et al., 2011*). The copper-diacetyl behavioral interaction requires ASH glutamatergic inhibition of AIA via the glutamate-gated chloride channel GLC-3. In this context, the integrated sensory state detected by AIA includes a repellent as well as an attractant, and a different combination of sensory inputs. The behavioral role of AIA activity is complex, but it is interesting that the combinations of sensory neurons that activate AIA are partly distinct from the roles of those sensory neurons in chemotaxis. For example, AWA is not necessary for chemotaxis to isoamyl alcohol, but has an important role in AIA activation by that odor, and the converse holds for ASK and diacetyl.

The existence of a complete wiring diagram for *C. elegans* poses a challenge: are there core computations performed by repeated circuit motifs? We speculate that the combination of inhibitory chemical synapses, excitatory gap junctions, and intrinsic bistability, all of which are present in multiple *C. elegans* neurons, could represent the core features of an AND-gate circuit motif. This possibility can be tested by further experiments.

# Materials and methods

## Key resources table

| Reagent type (species) or resource | Designation | Source or reference | Identifiers | Additional information |
|---|---|---|---|---|
| Strain, strain background (*Caenorhabditis elegans* N2, hermaphrodite) | *AWA::GCaMP2.2b* | DOI: 10.1016/j.celrep.2015.08.032 | ID_Bargmann Database:CX14647 | See *Figure 1*, *Figure 1—figure supplements 1*, *Figure 3*, *Figure 5—figure supplement 3* |
| Strain, strain background (*C. elegans* N2, hermaphrodite) | *AWA::Chrimson; AWA::GCaMP2.2b* | DOI: 10.1016/j.celrep.2015.08.032 | ID_Bargmann Database:CX16573 | See *Figure 1*, *Figure 1—figure supplements 1, 3*, *Figure 3*, *Figure 3—figure supplement 1* |
| Strain, strain background (*C. elegans* N2, hermaphrodite) | *AIA::GCaMP5A* | DOI: 10.1016/j.celrep.2015.08.032 | ID_Bargmann Database:CX15257 | See *Figure 1*, *Figure 1—figure supplements 2, 3*, *Figure 3—figure supplement 1*, *Figure 5*, *Figure 4—figure supplements 2*, *Figure 5—figure supplement 3*, *Figure 6*, *Figure 6—figure supplement 1* |
| Strain, strain background (*C. elegans* N2, hermaphrodite) | *AWA::Chrimson; AIA::GCaMP5A* | DOI: 10.1016/j.celrep.2015.08.032 | ID_Bargmann Database:CX16561 | See *Figure 1*, *Figure 1—figure supplements 2, 3*, *Figure 2—figure supplement 1*, *Figure 3*, *Figure 3—figure supplement 1*, *Figure 4*, *Figure 4—figure supplements 1, 2*, *Figure 5—figure supplement 3* |
| Strain, strain background (*C. elegans* N2, hermaphrodite) | *odr-7; AIA::GCaMP5A* | DOI: 10.1016/j.celrep.2015.08.032 | ID_Bargmann Database:CX16171 | See *Figure 2*, *Figure 5*, *Figure 4—figure supplement 2*, *Figure 5—figure supplement 3* |
| Strain, strain background (*C. elegans* N2, hermaphrodite) | *odr-10; AIA::GCaMP5A* | DOI: 10.1016/j.celrep.2015.08.032 | ID_Bargmann Database:CX16170 | See *Figure 2*, *Figure 4—figure supplement 2* |
| Strain, strain background (*C. elegans* N2, hermaphrodite) | *AWA::TeTx; AIA::GCaMP5A* | this paper | ID_Bargmann Database:CX16584 | See *Figure 2*, *Figure 4—figure supplement 2* |
| Strain, strain background (*C. elegans* N2, hermaphrodite) | *AWA::TeTx; AWA::Chrimson; AIA::GCaMP5A* | this paper | ID_Bargmann Database:CX17519 | See *Figure 2—figure supplement 1*, *Figure 4—figure supplement 2* |
| Strain, strain background (*C. elegans* N2, hermaphrodite) | *unc-7(e5); AIA::GCaMP5A* | this paper | ID_Bargmann Database:CX18039 | See *Figure 2*, *Figure 4—figure supplement 2* |
| Strain, strain background (*C. elegans* N2, hermaphrodite) | *unc-9(fc16); AIA::GCaMP5A* | this paper | ID_Bargmann Database:CX16980 | See *Figure 2* |
| Strain, strain background (*C. elegans* N2, hermaphrodite) | *unc-9 unc-7; AIA::GCaMP5A* | this paper | ID_Bargmann Database:CX16979 | See *Figure 2*, *Figure 2—figure supplement 1*, *Figure 4—figure supplement 2* |
| Strain, strain background (*C. elegans* N2, hermaphrodite) | *unc-9 unc-7; AWA,AIA::unc-9(-); AIA::GCaMP5A* | this paper | ID_Bargmann Database:CX18040 | See *Figure 2*, *Figure 2—figure supplement 1*, *Figure 4—figure supplement 2* |

*Continued on next page*

*Continued*

| Reagent type (species) or resource | Designation | Source or reference | Identifiers | Additional information |
|---|---|---|---|---|
| Strain, strain background (*C. elegans* N2, hermaphrodite) | *unc-9 unc-7; AWA,AIA::unc-9(+); AIA::GCaMP5A* | this paper | ID_Bargmann Database:CX18041 | See *Figure 2*, *Figure 2—figure supplement 1*, *Figure 4—figure supplement 2* |
| Strain, strain background (*C. elegans* N2, hermaphrodite) | *unc-7 unc-9; AWA::Chrimson; AIA::GCaMP5A* | this paper | ID_Bargmann Database:CX17320 | See *Figure 2*, *Figure 5—figure supplement 3* |
| Strain, strain background (*C. elegans* N2, hermaphrodite) | *AIA::Chrimson; AIA::GCaMP5A* | this paper | ID_Bargmann Database:CX17432 | See *Figure 2*, *Figure 2—figure supplement 1* |
| Strain, strain background (*C. elegans* N2, hermaphrodite) | *unc-7 unc-9; AIA::Chrimson; AIA::GCaMP5A* | this paper | ID_Bargmann Database:CX17895 | See *Figure 2* |
| Strain, strain background (*C. elegans* N2, hermaphrodite) | *AIA::Chrimson; AWA::GCaMP2.2b* | this paper | ID_Bargmann Database:CX17464 | See *Figure 2*, *Figure 2—figure supplement 1* |
| Strain, strain background (*C. elegans* N2, hermaphrodite) | *unc-7 unc-9; AIA::Chrimson; AWA::GCaMP2.2b* | this paper | ID_Bargmann Database:CX17897 | See *Figure 2* |
| Strain, strain background (*C. elegans* N2, hermaphrodite) | *unc-18; AIA::Chrimson; AIA::GCaMP5A* | this paper | ID_Bargmann Database:CX17584 | See *Figure 2— figure supplement 1* |
| Strain, strain background (*C. elegans* N2, hermaphrodite) | *unc-18; AIA::Chrimson; AWA::GCaMP2.2b* | this paper | ID_Bargmann Database:CX17640 | See *Figure 2— figure supplement 1* |
| Strain, strain background (*C. elegans* N2, hermaphrodite) | *unc-13; AIA::GCaMP5A* | this paper | ID_Bargmann Database:CX16591 | See *Figure 3*, *Figure 3—figure supplement 1*, *Figure 4—figure supplement 2* |
| Strain, strain background (*C. elegans* N2, hermaphrodite) | *unc-18(e234); AIA::GCaMP5A* | this paper | ID_Bargmann Database:CX16412 | See *Figure 3*, *Figure 3—figure supplement 1*, *Figure 4—figure supplement 2* |
| Strain, strain background (*C. elegans* N2, hermaphrodite) | *unc-13; AWA::Chrimson; AIA::GCaMP5A* | this paper | ID_Bargmann Database:CX16592 | See *Figure 3*, *Figure 4—figure supplement 2* |
| Strain, strain background (*C. elegans* N2, hermaphrodite) | *unc-18(e234); AWA::Chrimson; AIA::GCaMP5A* | this paper | ID_Bargmann Database:CX17158 | See *Figure 3*, *Figure 3—figure supplement 1*, *Figure 4*, *Figure 4—figure supplement 2* |
| Strain, strain background (*C. elegans* N2, hermaphrodite) | *unc-18(e81); AWA::Chrimson; AIA::GCaMP5A* | this paper | ID_Bargmann Database:CX17640 | See *Figure 3*, *Figure 3—figure supplement 1*, *Figure 4*, *Figure 4—figure supplement 2* |
| Strain, strain background (*C. elegans* N2, hermaphrodite) | *unc-13; AWA::Chrimson; AWA::GCaMP2.2b* | this paper | ID_Bargmann Database:CX17213 | See *Figure 3*, *Figure 3—figure supplement 1* |
| Strain, strain background (*C. elegans* N2, hermaphrodite) | *unc-31; AWA:: Chrimson; AIA::GCaMP5A* | this paper | ID_Bargmann Database:CX17319 | See *Figure 3— figure supplement 1* |

*Continued on next page*

*Continued*

| Reagent type (species) or resource | Designation | Source or reference | Identifiers | Additional information |
|---|---|---|---|---|
| Strain, strain background (*C. elegans* N2, hermaphrodite) | *eat-4-FRT; AWA::Chrimson; AIA::GCaMP5A* | this paper | ID_Bargmann Database:CX17714 | See *Figure 4*, *Figure 4—figure supplement 1*, *Figure 4—figure supplement 2* |
| Strain, strain background (*C. elegans* N2, hermaphrodite) | *eat-4-FRT; AWC,ASE,ASK, ASG::nFlippase; AWA::Chrimson; AIA::GCaMP5A* | this paper | ID_Bargmann Database:CX17679 | See *Figure 4*, *Figure 4—figure supplement 1*, *Figure 4—figure supplement 2* |
| Strain, strain background (*C. elegans* N2, hermaphrodite) | *eat-4-FRT; ASK::nFlippase; AWA::Chrimson; AIA::GCaMP5A* | this paper | ID_Bargmann Database:CX17722 | See *Figure 4*, *Figure 4—figure supplement 1*, *Figure 4—figure supplement 2* |
| Strain, strain background (*C. elegans* N2, hermaphrodite) | *eat-4-FRT; ASG::nFlippase; AWA::Chrimson; AIA::GCaMP5A* | this paper | ID_Bargmann Database:CX17892 | See *Figure 4*, *Figure 4—figure supplement 1*, *Figure 4—figure supplement 2* |
| Strain, strain background (*C. elegans* N2, hermaphrodite) | *eat-4-FRT; AWC::nFlippase; AWA::Chrimson; AIA::GCaMP5A* | this paper | ID_Bargmann Database:CX17611 | See *Figure 4*, *Figure 4—figure supplement 1*, *Figure 4—figure supplement 2* |
| Strain, strain background (*C. elegans* N2, hermaphrodite) | *eat-4-FRT; AWC,ASE::nFlippase; AWA::Chrimson; AIA::GCaMP5A* | this paper | ID_Bargmann Database:CX17723 | See *Figure 4*, *Figure 4—figure supplement 1*, *Figure 4—figure supplement 2* |
| Strain, strain background (*C. elegans* N2, hermaphrodite) | *che-1; AWA::Chrimson; AIA::GCaMP5A* | this paper | ID_Bargmann Database:CX17678 | See *Figure 4*, *Figure 4—figure supplement 1* |
| Strain, strain background (*C. elegans* N2, hermaphrodite) | *AWC,ASE,ASK, ASG::nFlippase; AWA::Chrimson; AIA::GCaMP5A* | this paper | ID_Bargmann Database:CX17866 | See *Figure 4*, *Figure 4—figure supplement 1*, *Figure 4—figure supplement 2* |
| Strain, strain background (*C. elegans* N2, hermaphrodite) | *unc-18; AWC,ASE::unc-18(+); AWA::Chrimson; AIA::GCaMP5A (line A)* | this paper | ID_Bargmann Database:CX17675 | See *Figure 4*, *Figure 4—figure supplement 1* |
| Strain, strain background (*C. elegans* N2, hermaphrodite) | *unc-18; AWC,ASE::unc-18(+); AWA::Chrimson; AIA::GCaMP5A (line B)* | this paper | ID_Bargmann Database:CX17676 | See *Figure 4*, *Figure 4—figure supplement 1* |
| Strain, strain background (*C. elegans* N2, hermaphrodite) | *unc-18; AWC,ASE::unc-18(-); AWA::Chrimson; AIA::GCaMP5A* | this paper | ID_Bargmann Database:CX17677 | See *Figure 4*, *Figure 4—figure supplement 1* |
| Strain, strain background (*C. elegans* N2, hermaphrodite) | *ASK::GCaMP5A* | DOI: 10.1016/j.neuron. 2019.01.053 | ID_Bargmann Database:CX17590 | See *Figure 5*, *Figure 5—figure supplement 1*, *Figure 5—figure supplement 3* |
| Strain, strain background (*C. elegans* N2, hermaphrodite) | *unc-18; ASK::GCaMP5A* | this paper | ID_Bargmann Database:CX17724 | See *Figure 5— figure supplement 1* |
| Strain, strain background (*C. elegans* N2, hermaphrodite) | *odr-10; ASK::GCaMP5A* | this paper | ID_Bargmann Database:CX17867 | See *Figure 5— figure supplement 1* |

*Continued*

| Reagent type (species) or resource | Designation | Source or reference | Identifiers | Additional information |
|---|---|---|---|---|
| Strain, strain background (*C. elegans* N2, hermaphrodite) | *AWC::GCaMP5A* | this paper | ID_Bargmann Database:CX17520 | See *Figure 5*, *Figure 5—figure supplement 1*, *Figure 5—figure supplement 3* |
| Strain, strain background (*C. elegans* N2, hermaphrodite) | *unc-18; AWC::GCaMP5A* | this paper | ID_Bargmann Database:CX17636 | See *Figure 5— figure supplement 1* |
| Strain, strain background (*C. elegans* N2, hermaphrodite) | *odr-10; AWC::GCaMP5A* | this paper | ID_Bargmann Database:CX17606 | See *Figure 5— figure supplement 1* |
| Strain, strain background (*C. elegans* N2, hermaphrodite) | *ASE::GCaMP3* | this paper | ID_Bargmann Database:CX14571 | See *Figure 5*, *Figure 5—figure supplement 1* |
| Strain, strain background (*C. elegans* N2, hermaphrodite) | *unc-18; ASE::GCaMP3* | this paper | ID_Bargmann Database:CX17638 | See *Figure 5*, *Figure 5—figure supplement 1* |
| Strain, strain background (*C. elegans* N2, hermaphrodite) | *odr-10; ASE::GCaMP3* | this paper | ID_Bargmann Database:CX16497 | See *Figure 5*, *Figure 5—figure supplement 1* |
| Strain, strain background (*C. elegans* N2, hermaphrodite) | *ASH::GCaMP3* | DOI: 10.1016/j.neuron.2013.11.020 | ID_Bargmann Database:CX10979 | See *Figure 5— figure supplement 1* |
| Strain, strain background (*C. elegans* N2, hermaphrodite) | *AWA::Chrimson; ASK::GCaMP5A* | this paper | ID_Bargmann Database:CX17751 | See *Figure 5— figure supplement 1* |
| Strain, strain background (*C. elegans* N2, hermaphrodite) | *AWA::Chrimson; AWC::GCaMP5A* | this paper | ID_Bargmann Database:CX17521 | See *Figure 5— figure supplement 1* |
| Strain, strain background (*C. elegans* N2, hermaphrodite) | *AWA::Chrimson; ASE::GCaMP3* | this paper | ID_Bargmann Database:CX17392 | See *Figure 5— figure supplement 1* |
| Strain, strain background (*C. elegans* N2, hermaphrodite) | *ceh-36; AIA::GCaMP5A* | DOI: 10.1016/j.celrep.2015.08.032 | ID_Bargmann Database:CX16169 | See *Figure 5*, *Figure 5—figure supplement 3* |
| Strain, strain background (*C. elegans* N2, hermaphrodite) | *odr-7 ceh-36; AIA::GCaMP5A* | DOI: 10.1016/j.celrep.2015.08.032 | | See *Figure 5*, *Figure 5—figure supplement 3* |
| Strain, strain background (*C. elegans* N2, hermaphrodite) | *ASK,AWA::GCaMP6s; AIA::GCaMP5A* | this paper | ID_Bargmann Database:CX18038 | See *Figure 5— figure supplement 2* |
| Strain, strain background (*C. elegans* N2, hermaphrodite) | *AIA::GFP* | this paper | ID_Bargmann Database:CX8293 | See *Figure 6*, *Figure 6—figure supplement 1* |
| Strain, strain background (*C. elegans* N2, hermaphrodite) | *AIA::GCaMP5A* | this paper | ID_Bargmann Database:CX16976 | See *Figure 6*, *Figure 6—figure supplement 1* |

*Continued on next page*

*Continued*

| Reagent type (species) or resource | Designation | Source or reference | Identifiers | Additional information |
|---|---|---|---|---|
| Chemical compound, drug | (-)-tetramisole hydrochloride | Sigma | L9756 | CAS 16595-80-5 |
| Chemical compound, drug | Polydimethylsiloxane (PDMS) | Sigma | 761036 | 9:1 base:curing agent, Sylgard 184 |
| Software, algorithm | ImageJ | ImageJ (http://imagej.nih.gov/ij/) | RRID:SCR_003070 | Version 1.52a |
| Software, algorithm | GraphPad Prism | GraphPad Prism (https://graphpad.com) | RRID:SCR_002798 | Version 8 |
| Software, algorithm | Matlab | MathWorks (https://www.mathworks.com/) | RRID:SCR_001622 | Versions R2013b and R2015a |
| Software, algorithm | Metamorph | Molecular Devices (https://www.moleculardevices.com) | RRID:SCR_002368 | Versions 7.7.6 and 7.7.8 |
| Software, algrorithm | analysis code | this paper | | See *Source code 1* |

## *C. elegans* growth

We used standard genetic and molecular techniques (*Brenner, 1974*). *C. elegans* strains were maintained at 22℃ on Nematode Growth Media (NGM; 51.3 mM NaCl, 1.7% agar, 0.25% peptone, 1 mM $CaCl_2$, 12.9 µM cholesterol, 1 mM $MgSO_4$, 25 mM $KPO_4$ at pH 6) plates seeded with LB-grown *Escherichia coli* OP50 bacteria. Animals had constant access to food for at least three generations prior to experiments. Experiments were performed on young adult hermaphrodites. All strains and full genotypes are listed in *Supplementary file 5*.

## Stimulus preparation

Stimulus solutions were freshly prepared each experimental day by serially diluting from a pure stock of diacetyl (2,3-butanedione; Sigma-Aldrich 11038, CAS 431-03-8; stored at 4℃) or isoamyl alcohol (EMD AX-1440–6, CAS 123-51-3; stored at 4℃), or by directly dissolving NaCl (Fisher Chemical S271-1, CAS 7647-14-5) into S Basal buffer (0.1 M NaCl, 5.74 mM $K_2HPO_4$ and 44.1 mM $KH_2PO_4$ at pH 6, 5 µg/ml cholesterol). *E. coli* OP50 conditioned medium was prepared by seeding 30 ml NGM buffer without agar or cholesterol with a colony of OP50 bacteria and shaking the culture at 37℃ overnight such that optical density was ~0.2 by the morning of the experiment. The OP50 culture was filtered (0.22 µm Millex GP) before the experiment and NGM was used as the buffer control for these experiments. All stimulus solutions contained 1 mM (-)-tetramisole hydrochloride (Sigma L9756, CAS 16595-80-5) to paralyze the body wall muscles and were stored in brown glass vials, except for those used to simultaneously image AIA, AWA, and ASK. For simultaneous imaging experiments, stimulus solutions did not contain (-)-tetramisole hydrochloride and were stored in 30 ml plastic syringes that had been separately soaked in ethanol or water overnight.

## Calcium imaging of individual neurons

The calcium imaging protocol was adapted from *Larsch et al. (2015)*. Animals expressing GCaMP were selected as L4 larvae the evening before the experiment and picked to new OP50 plates. For Chrimson experiments, L4 animals were transferred to freshly seeded plates of 5x concentrated OP50-seeded LB with or without 5 µM all-trans retinal (Sigma R2500; CAS 116-31-4) and housed overnight in complete darkness.

Before beginning the experiment, we selected animals for visible GCaMP fluorescence and gently washed them in S Basal buffer. We then loaded ~10 animals of two genotypes or conditions into separate arenas of a custom-fabricated two-arena polydimethylsiloxane (PDMS; Sigma 761036, made from 9:1 base:curing agent, Sylgard 184) imaging devices that had been de-gassed in a vacuum dessicator for at least 5 min. We chose ~10 animals per arena to maximize the number of

replicates within a condition or genotype without sacrificing proper flow. Animals were paralyzed in darkness for ~90 min in buffer + tetramisole before the start of the recording.

Experiments were performed on a Zeiss AxioObserver A1 inverted microscope fit with a 5x/0.25 NA Zeiss Fluar objective. A Hamamatsu Orca Flash 4 sCMOS camera was mounted to the microscope using a 0.63x c-mount adapter to increase field of view. We delivered 474 nm wavelength light with a Lumencor SOLA-LE lamp. We used Metamorph 7.7.6 software to control image acquisition and light pulsing in addition to rapid stimulus switching (National Instruments NI-DAQmx connected to an Automate Valvebank 8 II actuator that controls a solenoid valve), odor selection (Hamilton 8-way distribution valve), and activation of an external red LED for Chrimson stimulation (Mightex Precision LED Spot Light, 617 nm, PLS-0617–030 s; attached to Chroma ET605/50x filter to narrow band to $605 \pm 25$ nm). For Chrimson experiments, red light intensity was 15 mW/cm$^2$.

For odor-only calcium imaging experiments, we detected GCaMP signals after illumination with 165 mW/cm$^2$ 474 nm light, strobed at a 10 ms per 100 ms (10 fps) exposure duty cycle to reduce motion artefacts and GCaMP photobleaching.

For optogenetic experiments, we used a lower light intensity at 474 nm to reduce blue light activation of the Chrimson channel. To define conditions, AWA neurons expressing GCaMP and Chrimson were tested at 474 nm light intensities ranging from 15 mW/cm$^2$ with a 10% duty cycle to 165 mW/cm$^2$ with constant illumination (*Figure 1—figure supplement 1D*). Strobed (10% duty cycle) 40 mW/cm$^2$ light captured fluorescent GCaMP signals in the AIA neurite, and retained full AWA responses to 617 nm Chrimson illumination and odor stimuli (*Figure 1—figure supplement 1D*), although with a higher level of background noise than under strong illumination.

Animals received two pulses of the tested stimulus, and both pulses were pooled for analyses, with the exception of *Figure 1—figure supplements 1D–E* and *3F–R*. In experiments with multiple odor concentrations, we delivered odors in order of increasing concentration. In Chrimson experiments with diacetyl controls (*Figure 1—figure supplement 1D* and *Figure 1—figure supplement 3C*), or diacetyl experiments with NaCl controls (*Figure 5—figure supplement 1D and H*), the control was delivered last. To confirm proper odor flow, we delivered a pulse of fluorescein dye at the end of the experiment; assays in which flow was impeded were discarded.

Raw fluorescence values were measured using a custom ImageJ script from *Larsch et al. (2013)*, which measures the average intensity of a $4 \times 4$ pixel square and subtracts the local background intensity. For all sensory neurons, the square captured the soma; for AIA, it captured the middle of the neurite. Animals that moved too much for the tracking script, and animals with no visible AIA soma, were discarded. Occasionally, the tracking script inserted NaN (Not a Number) instead of a fluorescence intensity; pulses with NaN values were discarded. The tracking script is semi-automated to reduce experimenter bias. Each background-subtracted raw fluorescence trace was first normalized to generate $\Delta F/F_0$, where $F_0$ was the median of the 10 s (100 frames) before the odor pulse onset. Traces were then smoothed by five frames such that each frame $t$ represented the mean of $t$-2 frames to $t$+2 frames.

In a preliminary experiment, we delivered 28 AWA::Chrimson pulses to wild type animals, and 24 pulses to *unc-13(e51)* mutant animals (not included in analyses because we used a slightly different protocol). This experiment captured a statistically significant difference in cumulative response time profiles between wild type and *unc-13* AIA responses to AWA::Chrimson stimulation (Kolmogorov-Smirnov test, p: 0.007, D test statistic: 0.470). Guided by this result, all experiments reported here included at least 17 light or odor pulses (and typically >30) per genotype or condition.

We delivered two light or odor pulses per animal, tested 5–10 animals per PDMS device arena, tested 2–16 arenas per genotype or condition, and tested 2–10 genotypes or conditions per experimental block. The number of arenas per genotype or condition depended on the number of genotypes or conditions included in an experimental block, such that each genotype or condition was tested on at least two days with fresh odorant solutions or retinal plates, tested in both the upper and lower arenas of a PDMS device (see *Figure 1—figure supplement 1A*), and tested simultaneously with the control at least once. The n used for analysis refers to individual stimulus pulses. Statistical analyses include all genotypes or conditions in an experimental block, corrected for multiple comparisons that in some cases include genotypes or conditions we do not present. The genotypes or conditions included in an experimental block depended on the hypothesis we were testing. We included a wild type-to-*unc-18(e234)* comparison in several independent experimental blocks,

and the cumulative response time profiles for the separate and combined experimental blocks are shown in *Figure 3D*.

## Determining response latency times

For AWA, AIA, ASE, and ASH, a calcium trace was deemed a 'response' at the first frame $t$ at which the mean smoothed $\Delta F/F_0$ of $t$ to $t+12$ frames exceeded two standard deviations of the mean of the 10 s pre-stimulus $\Delta F/F_0$, and the mean time derivative of $t$ to $t+1$ frame exceeded one standard deviation of the mean time derivative of the 10 s pre-stimulus $\Delta F/F_0$. The threshold of 2 standard deviations of the pre-stimulus $\Delta F/F_0$ marks an inflection point in the number of traces categorized as 'responses' to buffer, in the absence of odor or Chrimson stimulus. This threshold was therefore the lowest, or most inclusive, threshold possible without including an excessive number of obvious non-responses, or false positives. The threshold of 2 standard deviation of the mean time derivative of the pre-stimulus $\Delta F/F_0$ marks an inflection point in the number of traces categorized as 'nonresponses' to each stimulus; that is, higher thresholds exclude responses. Therefore, this threshold is the highest threshold possible without excluding an excessive number of responses, or false negatives. Notably, at the threshold of 2 standard deviations of the pre-stimulus $\Delta F/F_0$, the number of responses are largely independent of the time derivative threshold, which we used to constrain the timing of the event. More details on thresholding procedures can be found in *Figure 1—figure supplement 2*.

To determine ASK and AWC response latency, but not for other calculations, each ASK and AWC calcium trace was scaled such that the minimum value was 0 and the maximum value was 1. The calcium trace was deemed a 'response' at the first frame $t$ at which the mean scaled $\Delta F/F_{max}$ of $t$ to $t+10$ frames was below two standard deviations of the mean of the 10 s pre-stimulus $\Delta F/F_0$, and either the time derivative of $t$ to $t+10$ frames was below 0.5 standard deviations, or the time derivative to $t$ to $t+5$ frames was below 1.15 standard deviations, of the mean of the 10 s pre-stimulus $\Delta F/F_0$.

To compare the variability of response latencies, we compared the cumulative response time profiles. We used the Kolmogorov-Smirnov test to compare these distributions since this test would capture both the latencies and probability of response. Although the figures show only 5 s of stimulus, the Kolmogorov-Smirnov test compared distributions for 10 s of stimulus. Details of each test, including the D test statistic, can be found in *Supplementary file 2*.

## Comparing AWA-to-AIA lag times

To calculate the mean lag between AWA and AIA responses (*Figure 1—figure supplement 3B*, *Figure 3G*), we subtracted the frame at which 50% of AWA neurons had responded from the frame at which 50% of AIA neurons had responded to a given stimulus. We performed this calculation 1000 times from randomly bootstrap-sampled populations that had the same $n$ as the true population, sampled with replacement. The standard deviation of the bootstrapped distribution was used as the standard error of the bootstrapped mean. The lag between simultaneously recorded AWA and AIA responses in 24 animals was calculated by subtracting each animal's AIA response initiation time from its AWA response initiation time (*Figure 5—figure supplement 2D*). These times were determined separately by eye by an individual blinded to the identity of the neuron and animal number.

## Measuring response magnitudes in sensory neurons

To calculate ASK, AWC, ASE, and ASH response magnitudes to a given stimulus, we subtracted the mean $\Delta F/F_0$ of 10 frames (1 s) prior to stimulus delivery from the mean $\Delta F/F_0$ of the final 10 frames within the odor pulse. Because AWA and AIA responses often adapt during the stimulus pulse, we defined AWA and AIA response magnitudes as the maximum $\Delta F/F_0$ within the 10 s stimulus pulse.

To compare AIA or AWA magnitudes between genotypes or conditions, we included only calcium traces that represent detectable responses as described in the Determining Latency Times section. To determine whether there was an appreciable ASK, AWC, or ASH response to a given stimulus, or AWA response to isoamyl alcohol, we included all calcium traces.

We statistically tested differences with either an ordinary one-way ANOVA with Dunnett's multiple comparisons test for experiments with more than two conditions, or an unpaired t-test for

experiments with only two conditions. For ASK, AWC, ASE, AWA, and ASH responses to diacetyl, isoamyl alcohol, or NaCl, we used responses to S Basal – S Basal buffer switches as the control. For ASK, AWC, and ASE responses to AWA::Chrimson stimulation in *Figure 5—figure supplement 1O*, we used a paired t-test to compare response magnitudes to the change in $\Delta F/F_0$ within a similar time window prior to the light pulse. Details of each test can be found in *Supplementary file 3*.

### Calculating rise times

To calculate rise times of AIA responses, we included only calcium traces that represent detectable responses as described in the Determining Latency Times section. We calculated the rise time by subtracting the time at which AIA reached 33% of its peak magnitude from the time at which AIA reached 66% of its peak magnitude for each response. We compared rise times to various stimuli using an ordinary one-way ANOVA with Dunnett's multiple comparisons test (three comparisons to AWA::Chrimson for *Figure 1—figure supplement 2D*, three comparisons to 1.15 µM diacetyl for *Figure 5—figure supplement 3M*, and three comparisons to wild type for *Figure 4—figure supplement 2M*).

### Simultaneous calcium imaging of multiple neurons

Animals expressing soluble GCaMP5A in AIA, and nuclear GCaMP6s in ASK, AWA, and several other sensory neurons, were selected as L4s the evening before the experiment and transferred to fresh bacterial plates overnight. Adult animals were transferred to an unseeded agar plate, then transferred into S Basal buffer containing 1 mM (-)-tetramisole hydrochloride as a paralytic agent. After 10 min, individual animals were loaded into a custom-fabricated PDMS microfluidic chamber in S Basal buffer, and either 11.5 nM or 1.15 µM diacetyl was delivered to the animal's nose under controlled conditions (*Chronis et al., 2007*). Fluorescence changes were monitored on a Zeiss Axiovert 100TV wide-field inverted microscope fitted with a 40x/1.3 Zeiss Plan Apochromat oil objective. Metamorph 7.7.8.0 software controlled stimulus switching and image acquisition through an Andor iXon+ EMCCD camera (as described above, Calcium Imaging of Individual Neurons). Light was delivered with a Lumencor SOLA-LE lamp, passed through a 1.3 ND filter, and narrowed to 484–492 nm using a CHROMA 49904-ET Laser Bandpass filter set. Illumination was constant and at full power, which may have elicited some light responses in ASK.

We used ImageJ's 'Align slices in stack' plugin to correct for small amounts of motion. We then measured fluorescence values by selecting regions of interest that captured the AIA neurite, AWA nucleus, and ASK nucleus without background subtraction. We scaled fluorescence values separately for each neuron from 0 to 1 for visualization purposes. Many videos were discarded because fluorescence from other neurons prevented measurement of our neurons of interest, because our neurons of interest could not be captured in the same focal plane, or because the animal rotated during the video.

### Electrophysiological recordings

Electrophysiological recording was performed as previously described (*Liu et al., 2018*). Briefly, an adult animal was immobilized on a Sylgard-coated (Sylgard 184, Dow Corning) glass coverslip in a small drop of DPBS (D8537; Sigma) by applying a cyanoacrylate adhesive (Vetbond tissue adhesive; 3M) along the dorsal side of the head region. A puncture in the cuticle away from the head was made to relieve hydrostatic pressure. A small longitudinal incision was then made using a diamond dissecting blade (Type M-DL 72029 L; EMS) between two pharyngeal bulbs along the glue line. The cuticle flap was folded back and glued to the coverslip with GLUture Topical Adhesive (Abbott Laboratories), exposing the nerve ring. The gluing and dissection were performed under an Olympus SZX16 stereomicroscope equipped with a 1X Plan Apochromat objective and widefield 10x eyepieces. The coverslip with the dissected preparation was then placed into a custom-made open recording chamber (~1.5 ml volume) and treated with 1 mg/ml collagenase (type IV; Sigma) for ~10 s by hand pipetting. The recording chamber was subsequently perfused with the desired extracellular solution using a custom-made gravity-feed perfusion system for ~10 ml. All experiments were performed with the bath at room temperature. We used an upright microscope (Axio Examiner; Carl Zeiss, Inc) equipped with a 40x water immersion lens and 16x eyepieces to view the preparation. AIA neurons were identified using either GFP or GCaMP5A fluorescent markers. We made

electrodes by using a laser pipette puller (P-2000; Sutter Instruments) to pre-pull borosilicate glass pipettes (BF100-58-10; Sutter Instruments) with resistance (RE) of 15–20 MΩ and back-filling them with the desired intracellular solutions. We used a motorized micromanipulator (PatchStar Micromanipulator; Scientifica) to control the electrodes. The pipette solution was (all concentrations in mM): [K-gluconate 115; KCl 15; KOH 10; $MgCl_2$ 5; $CaCl_2$ 0.1; $Na_2ATP$ 5; NaGTP 0.5; Na-cGMP 0.5; cAMP 0.5; BAPTA 1; Hepes 10; Sucrose 50], with pH adjusted with KOH to 7.2, osmolarity 320–330 mOsm. The standard extracellular solution was: [NaCl 140; NaOH 5; KCL 5; $CaCl_2$ 2; $MgCl_2$ 5; Sucrose 15; Hepes 15; Dextrose 25], with pH adjusted with NaOH to 7.3, osmolarity 330–340 mOsm. Liquid junction potentials were calculated and corrected before recording. Whole-cell current clamp and voltage clamp experiments were conducted on an EPC-10 amplifier (EPC-10 USB; Heka) using PatchMaster software (Heka). Two-component capacitive compensation was optimized at rest, and series resistance was compensated to 50%. Analog data were filtered at 2 kHz and digitized at 10 kHz or 50 kHz. For quality control, only patch clamps with a seal resistance above 1 GΩ and uncompensated series resistance below 100 MΩ were accepted for further analysis (seal resistance ranged from 2 to 10 GΩ; series resistance ranged from 17 to 95 MΩ for most accepted recordings). For the 1 pA/step experiment, a short −10 pA hyperpolarizing pre-pulse was added to each current injection step in the stimulation protocol to reset membrane potential to rest. Voltage or current measurement and data analysis were conducted using Fitmaster (Heka) and exported to OriginProt (OriginLab) for graphing. To reduce file size, electrophysiological recording traces shown in figures were re-sampled at 1 kHz by averaging adjacent data points.

## Simultaneous electrophysiology and calcium imaging

The electrophysiology recording setup was as described above. GCaMP fluorescence in AIA neurons expressing soluble GCaMP5A was captured with a CoolSNAP HQ2 Camera (Photometrics) controlled by MetaMorph software. The onset of imaging acquisition and electrophysiology recording was synchronized by a TTL signal sent through Patchmaster. We illuminated the sample with a SpectraX Lumencor solid-state light source. Patch-clamping was performed under DIC illumination, and the sample was switched to fluorescent illumination once a whole-cell recording configuration was formed to begin simultaneous recording. Imaging was set at 20 MHz, binning at 2 and 50 fps. To quantify GCaMP fluorescence, we used ImageJ to capture the mean fluorescence of a hand-selected area around the AIA neurite. We bleach-corrected the resulting trace by fitting a line to the non-stimulus portions of the calcium trace, then subtracted and subsequently dividing the raw trace by the fitted line to achieve the $\Delta F/F_0$.

## Acknowledgements

We thank Philip Kidd, Sagi Levy, Aylesse Sordillo, Elias Scheer, Du Cheng, Audrey Harnagel, Alejandro López-Cruz, Likui Feng, James Lee, Javier Marquina-Solis, Andrew Gordus, Andrew Leifer, Vanessa Ruta, and Shai Shaham for thoughtful discussions and comments on the manuscript. We thank Mei Zhen for the *unc-7 unc-9* double mutant strain. This work was supported by the Howard Hughes Medical Institute, of which CIB was an investigator, by a grant from the Kavli Neural Systems Institute to QL, and by the Chan Zuckerberg Initiative.

## Additional information

### Funding

| Funder | Author |
| --- | --- |
| Chan Zuckerberg Initiative | May Dobosiewicz<br>Qiang Liu<br>Cornelia I Bargmann |
| Howard Hughes Medical Institute | May Dobosiewicz<br>Qiang Liu<br>Cornelia I Bargmann |
| Kavli Neural Systems Institute | Qiang Liu |

The funders had no role in study design, data collection and interpretation, or the decision to submit the work for publication.

## Author contributions
May Dobosiewicz, Conceptualization, Formal analysis, Investigation, Writing—original draft, Writing—review and editing; Qiang Liu, Investigation, Writing—review and editing; Cornelia I Bargmann, Conceptualization, Supervision, Funding acquisition, Writing—review and editing

## Author ORCIDs
Cornelia I Bargmann (iD) https://orcid.org/0000-0002-8484-0618

## Decision letter and Author response
Decision letter https://doi.org/10.7554/eLife.50566.sa1
Author response https://doi.org/10.7554/eLife.50566.sa2

## Additional files

### Supplementary files
• Source code 1. Source code for data analysis.

• Supplementary file 1. Presynaptic partners of AIA.Both *Chen et al. (2006)* and *Cook et al. (2019)* are based on same collection of serial-section electron micrographs from *White et al. (1986)*. Neurotransmitter information is based on *Pereira et al. (2015)*.

• Supplementary file 2. Cumulative response time profiles.Kolmogorov-Smirnov test statistics and sample sizes for all cumulative response time profiles presented, calculated for full 10 s stimulus pulse. Italics indicate non-WT genetic backgrounds. D test represents the maximum effect size across the distributions. p-values below 0.05 are bolded for emphasis.

• Supplementary file 3. Calcium response magnitude comparisons.Magnitudes of responses to various stimuli, with either an unpaired t-test (if the number of comparisons is one) or an ordinary one-way ANOVA with Dunnett's multiple comparisons test (if the number of comparisons exceeds one); * indicates paired t-test. Bolded genotype or stimulus indicates the control group used for comparisons. Italics indicate non-wildtype genetic background. p-values below 0.05 are bolded for emphasis.

• Supplementary file 4. Calcium rise time comparisons.Rise times ($t_{66}$-$t_{33}$) of responses to various stimuli, with either an ordinary one-way ANOVA with Dunnett's multiple comparisons test. Bolded genotype or stimulus indicates the control group used for comparisons. Italics indicate non-wildtype genetic background.

• Supplementary file 5. Strain list.

• Transparent reporting form

### Data availability
All data generated or analyzed during this study, including source data, are included in the manuscript and supporting files.

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
