## [Decision Letter]

**Acceptance summary:**

In order to evaluate their environment, animals can process information via multiple sensory channels. The basic computations that lead to salient and behaviorally relevant neuronal representations, such as valence of an input, are an active area of research. The authors describe a *C. elegans* interneuron class, termed AIA, that receives information about attractive odors from multiple pre-synaptic primary sensory neurons. The sensory neuron AWA, which is activated by attractive odors, excites AIA via electrical signaling, while a set of sensory neurons that are inhibited by attractive odors, inhibit AIA via glutamatergic chemical synapses. This circuit motif performs an AND-gate computation during which concomitant excitation and disinhibition ensures reliable positive valence responses in AIA in the presence of low odor concentrations. In support of their model, the authors find that AIA exhibits non-linear bimodal voltage responses to input currents, suggesting a cellular mechanism for this computation. The computational motif described here perhaps repeats in the *C. elegans* nervous system; moreover, it is likely implemented in a similar way in circuits of animals with larger nervous systems.

**Decision letter after peer review:**

Thank you for submitting your article "Reliability of an interneuron response depends on an integrated sensory state" for consideration by *eLife*. Your article has been reviewed by three peer reviewers, one of whom is a member of our Board of Reviewing Editors, and the evaluation has been overseen by Catherine Dulac as the Senior Editor. The reviewers have opted to remain anonymous.

The reviewers have discussed the reviews with one another and the Reviewing Editor has drafted this decision to help you prepare a revised submission.

In this manuscript, Dobosiewicz and Bargmann use calcium imaging and genetic manipulations to examine the integration of sensory information by AIA interneurons in *C. elegans*. By comparing odor- and optogenetically- driven responses from upstream neurons, the authors find that input from a single sensory neuron (AWA) produces a slow and unreliable response, while coincident input from multiple sensory neurons produces more robust and reliable responses. The authors find that AWA provides excitatory input to AIA through electrical synapses, while ASK and AWC provide disinhibitory input through glutamatergic synapses. The authors conclude that integration onto AIA functions as an "AND-gate" and is involved in the selective filtering of sensory information for behavior.

A) Major concern:

The AND gate logic and filtering function seem largely speculative at the moment and more experiments are required to support this claim. The reviewers agree that summation at the level of AIA as opposed to a non-linear AND computation could equally well explain your observations. We understand that reviewer #3 suggestion to perform simultaneous imaging in AIA and the sensory neurons is perhaps out of reach within 2 Months of revisions due to the low resolution of your experimental setup. But we find it necessary that you perform additional convincing experiments that could address this issue in an equivalent manner (for example, by co-stimulating AIA with a specific ASK ligand, or voltage imaging).

Reviewer #3 also is concerned by the lack of explanation what causes the variability in AIA and that you cannot exclude effects from network states, like in Gordus et al. See also reviewer #1, comment (6). While we find it not necessary to address this concern with new experiments, the possible sources of variability should be better discussed.

Please pay also particular attention to rule out reviewer #3's concern that some of your results and conclusions are simply due to the thresholding procedures. A better explanation and a quantitative assessment of how certain parameters were chosen and that your choice does not skew the results is important.

B) Other essential revisions:

Reviewer #1:

1) The main experiment has the caveat that Chrimson is not only very sensitive but also has a long excitation tail extending to the wavelength (474nm) used for GCaMP imaging. Other experts in the field struggle with this problem. It is likely that GCaMP excitation light leads to basal tonic activation of AWA and perhaps habituation of signaling pathways parallel or downstream of Ca^++^. As stated in their Materials and methods, to prevent AWA activation, the authors use low 447nm light levels. In addition, a peculiar 474nm light-pulsing protocol (10 ms, every 100 ms) is used. I assume that these conditions have been somehow optimized to prevent AWA::Chrismon activation but a systematic demonstration of this is missing here as well as in previous literature. The authors should report in detail how they optimized and scrutinized their light stimulation protocol. This would be needed for the community to reproduce these data, and also extremely useful for other combined optogenetics/imaging experiments.

2) A crucial element of their model is that AWA signals unidirectionally via a gap junction to AIA. While inhibitory glutamatergic signaling to AIA via the other sensory neurons has been characterized, the support for AWA-AIA gap junction signaling is rather thin here.

2a) Conclusions made with the AWA::TeTx strain are based on a negative result, therefore some validation should be provided that this strain is effective at all. For example, AWA sends a relative high number of synapses to AIZ. Does optogenetic activation of AWA affect AIZ activity and is this altered in the AWA::TeTx strain?

2b) UNC-7 and UNC-9 are broadly expressed in the nervous system and are implicated in many functions. Therefore, indirect effects cannot be excluded.

- Is there no phenotype in single mutants?

- This concern could be addressed by transgenic rescue experiments in AWA and AIA.

- In addition, or alternatively, the authors have reported previously a tool for cell-class selective removal of *unc-9*, which could be used for AIA and AWA.

3) Figure 6. A-B: n-numbers are not listed in legend or table

The data would be more informative if the authors also showed the cumulative response probabilities of AWC, AWA and ASK as well as the AIA response delay time for WT.

4) The authors discuss that the function of this AND-gate computation might be "an integrative step that may filter out environmental noise". Is this consistent with the result in Figure 3A?: based on the authors model, we should assume that the AND-gate computation is absent in the synaptic transmission mutants. Here, I would expect unreliable responses of AIA to the weak AWA stimulation via 11.6nM dia (see Figure 1A), as opposed to when AWA gets strongly activated by Chrimson. This seeming contradiction and the functional relevance of the mechanism could be better discussed.

5) The authors discuss that AIA primarily is modulated by stimulus versus motor state. Although their wording is careful ("AIA activity is more closely coupled to sensory state, and less to motor state,"), a reader less acquainted with the literature could be misled. Currently, I don't see evidence for this statement. In contrast, in freely moving imaging studies Laurent et al., 2015, showed that AIA is indeed modulated by motor state; this paper should be cited and the implications to the current study should be discussed.

Reviewer #3:

1) Measures of response reliability and kinetics and interpretation of AIA integration as an AND-gate (But see our comments Major concern (A) above.)

The conclusion that AIA performs an "AND" gate is based in large part on data in Figure 1 showing that AWA:Chrimson produces smaller, more delayed and less likely responses in AIA than 115 nM diacetyl, however that response kinetics, when there is a response, are similar. However these two pieces of data seem to be in conflict. In the heat maps in Figure 1C (last column) there are clearly some responses with very slow kinetics. The fact that the two curves in Figure 1F are similar (though not the same) is likely an artefact of the thresholding used to determine that there was a response. The same is likely true in Figure 3D, which likewise shows an average of thresholded responses, while the heat maps in 3B show responses with variable latencies.

More generally, I think the authors should try to provide more experimental insight into the source of the observed variability. Some measures of variability and regression on expression levels of Chrimson and GCaMP are shown in the supplement to Figure 1, but only a subset of trials are ever shown in any of the figures. Do responses change in any systematic way over the time of the experiment? Are there slow fluctuations in response magnitude? Are responses correlated on some timescale? What do the authors think is the primary source of the variability in responses? Is it arising at the level of synaptic integration, or calcium channel activation? Is there a contribution of motor state variability to responses as in AIB, even if smaller?

Finally, I think conclusions about integration in AIA would be greatly strengthened by showing AIA responses as a function of simultaneously measured responses in AWA or ASK/AWC. Since simultaneous imaging experiments are mentioned in Figure 5, it seems like this data would be possible to obtain. Plotting AIA response magnitude on a trial-by-trial basis as a function of AWA/ASK/AWC activation on that trial would allow the authors to explicitly test and accept or reject the hypothesis that the integration between these sensory inputs is non-linear, versus being linear with some correlated or uncorrelated noise source either up or downstream. I think these analyses and an explicit model are important to draw the conclusions made here.

2) Temporal filtering by AIA

A second, more minor, conclusion drawn in the Discussion is that AIA integration serves to filter out transient or noisy odor stimuli. However, this idea is not explicitly tested in the manuscript. Odor dynamics can be challenging to control but can be done with proportional valves and measured by photo-ionization detector. Chrimson activation can be reliably controlled in time. If the authors wish to draw conclusions about temporal filtering or noise rejection in AIA I think they need to test this idea by explicitly varying the timecourse of odor or the level of background noise and measuring responses.

Reviewer #2:

1) Subsection “Calcium Imaging”, fourth paragraph: Both pulses were pooled for analyses.

For both heatmaps in Figure 1C last panel (AWA:Chr) and Figure 1—figure supplement 1E, it appears that probability of AIA response is <50% (with varying latencies) given each row is a single trial (n= 569). Number of animals tested (Figure 1—figure supplement 1J) are 282 where 35% respond to both AWA:Chr pulses, 28% to 1st only, and 16% to 2nd only. What is getting pooled in Figure 1? Only traces which respond to both pulses?

2) The mean trace for the AWA:Chr stimulus is similar to 115 nM diacetyl, but the trials seem more variable, including a number of non-responders. Are these most likely expression differences, or something about Chrimson dynamics? Can you be sure the low-level of 474 nm blue light used to image GCaMP prior to red light stimuluation does not have an effect on Chrimson? Did you try stimulating only the dendrite / sensory ending of AWA?

3) Did you attempt to image AWA and AIA at the same time, or is this precluded by the need to image the AIA process?

4) *unc-13* mutants as well another transgene of *unc-18* seem to have no effect on AIA response latency at 1.15uM diacetyl. Though this is mentioned in the second paragraph of the subsection “Chemical synapses inhibit AIA”, the rationale is not clearly explained. Is the reliability of the response also dosage dependent? Only data related to latency are shown in Figure 3A and Figure 3—figure supplement 1A, heatmaps which can indicate reliability are missing.

5) As in comment 2, only response time profiles are shown in Figure 4, indicating differences in latency, but heatmaps should be added as supplementary data to show trial-to-trial variability in responses.

6) At 11.5nM diacetyl, AWA is activated and ASK is inhibited, and this leads to moderate reliability of AIA response. At 1.15uM dia, AWA is activated with high fidelity, ASE is activated; ASK and AWC are inhibited and this leads to disinhibition of AIA and high reliability in its response. Do glutamate levels or different channel expression regulate the magnitude of disinhibition and in turn affect the reliability? ASK seems to be a key candidate tuning this reliability, as inhibition of ASK alone (Figure 4D) has a significant effect on AIA response reliability.

7) Glutamate release from ASE (with AWC::Figure 4I) can inhibit AIA activation. However, 1.15uM dia activates ASE, yet increases AIA reliability. This is not well clarified in the manuscript, and a model figure and extended discussion would help in this regard.

8) Were the *unc-9* and *unc-7* mutants tested individually to identify which of the innexins are required for AWA-AIA communication?

---

## [Author Response]

A) Major concern:The AND gate logic and filtering function seem largely speculative at the moment and more experiments are required to support this claim. The reviewers agree that summation at the level of AIA as opposed to a non-linear AND computation could equally well explain your observations. We understand that reviewer #3 suggestion to perform simultaneous imaging in AIA and the sensory neurons is perhaps out of reach within 2 Months of revisions due to the low resolution of your experimental setup. But we find it necessary that you perform additional convincing experiments that could address this issue in an equivalent manner (for example, by co-stimulating AIA with a specific ASK ligand, or voltage imaging).

We have developed the argument for the AND-gate logic further through (1) further documentation of the calcium imaging and its analysis to address the technical concerns raised by reviewer 3 (expanded Figure 1 and Figure 1—figure supplements 1-3, showing robustness of experimental and analytical methods) (2) inclusion of a few simultaneous multi-neuron imaging experiments consistent with the conclusion, although we note that the number is small (Figure 5—figure supplement 2, new data), and (3) inclusion of electrophysiological analysis of AIA neurons that demonstrates a non-linear, bimodal current-voltage relationship, and provides a mechanistic explanation for the non-linearity of AIA calcium responses to sensory inputs (Figure 6, new data).

Reviewer #3 also is concerned by the lack of explanation what causes the variability in AIA and that you cannot exclude effects from network states, like in Gordus et al. See also reviewer #1, comment (6). While we find it not necessary to address this concern with new experiments, the possible sources of variability should be better discussed.

In the Discussion, we expand on the statement that "AIA activity is more closely coupled to sensory state, and less to motor state" than AIB interneurons (reviewer 1). In support of this statement, we point out that AIA interneurons respond to >90% of strong odor stimuli in individual trials, whereas AIB interneurons, respond to <60% of the stimuli. We note that network states could explain the remaining 10% of the variability, as well as variability in response magnitude and duration, which are not the focus of this work. In addition, we note that in the wiring diagram, AIA has three times as many synaptic inputs from sensory neurons as from interneurons, whereas AIB has equal synaptic input from sensory neurons and interneurons.

Please pay also particular attention to rule out reviewer #3's concern that some of your results and conclusions are simply due to the thresholding procedures. A better explanation and a quantitative assessment of how certain parameters were chosen and that your choice does not skew the results is important.B) Other essential revisions:Reviewer #1:1) The main experiment has the caveat that Chrimson is not only very sensitive but also has a long excitation tail extending to the wavelength (474nm) used for GCaMP imaging. Other experts in the field struggle with this problem. It is likely that GCaMP excitation light leads to basal tonic activation of AWA and perhaps habituation of signaling pathways parallel or downstream of Ca^++^. As stated in their Materials and methods, to prevent AWA activation, the authors use low 447nm light levels. In addition, a peculiar 474nm light-pulsing protocol (10 ms, every 100 ms) is used. I assume that these conditions have been somehow optimized to prevent AWA::Chrismon activation but a systematic demonstration of this is missing here as well as in previous literature. The authors should report in detail how they optimized and scrutinized their light stimulation protocol. This would be needed for the community to reproduce these data, and also extremely useful for other combined optogenetics/imaging experiments.

Fair point, so did we, and we are happy to help the community as the reviewer tactfully suggests. In the new Figure 1—figure supplement 1, we show the methods development for this experiment. We calibrated conditions by expressing GCaMP and Chrimson in the same cell, AWA. Under strong illumination of GCaMP at 474 nm, we did not see fluorescence increases to longer-wavelength 617 nm Chrimson stimuli, presumably because of direct Chrimson excitation at 474 nm. Lower illumination levels at 474 nm resulted in a transient GCaMP response with a return to a baseline that allowed a subsequent Chrimson response to 617 nm light. We chose a light level at 474 nm that minimized the initial transient response and preserved a full AWA response to 617 nm light (left panels) and diacetyl odor (right panels). As stated in the legend to Figure 1—figure supplement 1, we used a 10 ms/100 ms duty cycle for 474 nm illumination, here and elsewhere because (a) strobing reduces motion artefacts and (b) the duty cycle minimizes GCaMP photobleaching during long-term imaging.

2) A crucial element of their model is that AWA signals unidirectionally via a gap junction to AIA. While inhibitory glutamatergic signaling to AIA via the other sensory neurons has been characterized, the support for AWA-AIA gap junction signaling is rather thin here.2a) Conclusions made with the AWA::TeTx strain are based on a negative result, therefore some validation should be provided that this strain is effective at all. For example, AWA sends a relative high number of synapses to AIZ. Does optogenetic activation of AWA affect AIZ activity and is this altered in the AWA::TeTx strain?2b) UNC-7 and UNC-9 are broadly expressed in the nervous system and are implicated in many functions. Therefore, indirect effects cannot be excluded.- Is there no phenotype in single mutants?- This concern could be addressed by transgenic rescue experiments in AWA and AIA.- In addition, or alternatively, the authors have reported previously a tool for cell-class selective removal of unc-9, which could be used for AIA and AWA.

We considered this experiment to be a confirmation of the classical White EM analysis rather than our own result, but we have addressed the reviewers’ points through discussion and the addition of further data. (2a) Validation of Tetanus toxin. Tetanus toxin has been effective in *C. elegans* in our hands and others in the past, and its target confirmed through the use of an uncleavable synaptobrevin (Macosko et al., 2009). We cite these results, and further elaborate that the Tetanus toxin experiment (a negative result) is supported by the subsequent demonstration that synaptic transmission mutants in *unc-13* and *unc-18* have enhanced, not diminished, signaling between AWA and AIA (Figure 3). (2b) We have added experiments showing that expression of the gap junction subunit *unc-9* in AIA and AWA (but not a point-mutated inactive version of *unc-9)* rescues the calcium imaging defect in an *unc-7 unc-9* double mutant (Figure 2F), and include the data showing that single mutants do not have a defect (Figure 2D).

3) Figure 6A-B: n-numbers are not listed in legend or table. The data would be more informative if the authors also showed the cumulative response probabilities of AWC, AWA and ASK as well as the AIA response delay time for WT.

We have added n values to the figure legend of Figure 5. The cumulative response profiles of sensory neurons and AIA to isoamyl alcohol are provided in Figure 5 for AIA and Figure 5—figure supplement 3 for ASK, AWA, and AWC.

4) The authors discuss that the function of this AND-gate computation might be "an integrative step that may filter out environmental noise". Is this consistent with the result in Figure 3A?: based on the authors model, we should assume that the AND-gate computation is absent in the synaptic transmission mutants. Here, I would expect unreliable responses of AIA to the weak AWA stimulation via 11.6nM dia (see Figure 1A), as opposed to when AWA gets strongly activated by Chrimson. This seeming contradiction and the functional relevance of the mechanism could be better discussed.

The noise filtering model has been de-emphasized in the Discussion. However, we do not agree that there is a contradiction between the model and the synaptic transmission mutant data. In our model, the AND-gate computation still applies in the synaptic transmission mutants, which have one condition of the AND-gate chronically fulfilled (no glutamate inhibition), leaving AWA activation (the second condition) largely in control of whether AIA responds. Our model predicts that the background of AIA spontaneous activity would increase in the synaptic transmission mutants, whereas the reviewer suggests that the reliability of the evoked signal would decrease. We believe the reviewer’s question may reflect a lack of clarity in our writing, so we have added a model figure making this point to Figure 6.

5) The authors discuss that AIA primarily is modulated by stimulus versus motor state. Although their wording is careful ("AIA activity is more closely coupled to sensory state, and less to motor state,"), a reader less acquainted with the literature could be misled. Currently, I don't see evidence for this statement. In contrast, in freely moving imaging studies Laurent et al., 2015, showed that AIA is indeed modulated by motor state; this paper should be cited and the implications to the current study should be discussed.

In support of this statement, we expand our description of the different results obtained with AIA interneurons, which respond to >90% of odor stimuli in individual trials, and the AIB interneurons, which respond to <60% of odor stimuli. We note that network states could explain the remaining 10% of the variability, as well as variability in response to magnitude and duration, which are not the focus of this work. We added a citation to the free-moving AIA imaging work from Laurent et al., but note that the relationship to behavior is complex; the authors of that paper take a light touch in their description of the one AIA trace that is shown (“careful analysis showed that [calcium changes] usually coincided with a switch in the direction of travel, and never lasted more than a few seconds”). The same paper showed much clearer evidence for behavior-correlated activity of AIB, AVA, and AVB interneurons.

Reviewer #3:1) Measures of response reliability and kinetics and interpretation of AIA integration as an AND-gate (But see our comments Major concern (A) above.)The conclusion that AIA performs an "AND" gate is based in large part on data in Figure 1 showing that AWA:Chrimson produces smaller, more delayed and less likely responses in AIA than 115 nM diacetyl, however that response kinetics, when there is a response, are similar. However these two pieces of data seem to be in conflict. In the heat maps in Figure 1C (last column) there are clearly some responses with very slow kinetics. The fact that the two curves in Figure 1F are similar (though not the same) is likely an artefact of the thresholding used to determine that there was a response. The same is likely true in Figure 3D, which likewise shows an average of thresholded responses, while the heat maps in 3B show responses with variable latencies.More generally, I think the authors should try to provide more experimental insight into the source of the observed variability. Some measures of variability and regression on expression levels of Chrimson and GCaMP are shown in the supplement to Figure 1, but only a subset of trials are ever shown in any of the figures. Do responses change in any systematic way over the time of the experiment? Are there slow fluctuations in response magnitude? Are responses correlated on some timescale? What do the authors think is the primary source of the variability in responses? Is it arising at the level of synaptic integration, or calcium channel activation? Is there a contribution of motor state variability to responses as in AIB, even if smaller?Finally, I think conclusions about integration in AIA would be greatly strengthened by showing AIA responses as a function of simultaneously measured responses in AWA or ASK/AWC. Since simultaneous imaging experiments are mentioned in Figure 5, it seems like this data would be possible to obtain. Plotting AIA response magnitude on a trial-by-trial basis as a function of AWA/ASK/AWC activation on that trial would allow the authors to explicitly test and accept or reject the hypothesis that the integration between these sensory inputs is non-linear, versus being linear with some correlated or uncorrelated noise source either up or downstream. I think these analyses and an explicit model are important to draw the conclusions made here.

We addressed the reviewer’s questions about the robustness of the analysis methods by re-formulating Figure 1, showing more of the analysis methods in an additional supplementary figure, and addressing them directly in that section of the Results. To expand:

a) Thresholding. We have documented the robustness of the decisions both by including more primary data in Figure 1 and by adding a supplementary figure (Figure 1—figure supplement 2) documenting thresholding and the robustness of the result. We identify a broad range of thresholds of signal magnitude and time derivative that yield a consistent set of results showing similar AIA responses to different stimuli. Importantly, we chose a set of thresholds in which response probability and latency are not sensitive to 2-fold increases or decreases in any parameter. The fact that peak response magnitude was still sensitive to these parameters is one of the reasons that we chose not to emphasize response magnitude in this study. The reviewer raises questions about the responses being “smaller, more delayed, and less likely.” The responses are not smaller in magnitude across stimuli, so we think the reviewer is talking about response duration, which does vary. What we want to emphasize is the sharp onset kinetics, not the duration of the stimulus; in fact, due to the obvious variability in duration and decay, it doesn’t make sense to average them. To bring that primary result out more, we added Figure 1E, showing representative single traces selected at random from the full set of traces, in the same latency order as the heatmap traces, demonstrating the sharp characteristic onset and variable duration. Figure 1E should also help provide visual evidence that the signal is high compared to the background. Figure 1—figure supplement 2, which shows an additional 150 individual responses, should also increase confidence in the basic robustness of the response onset. We moved all other aligned /averaged responses to Figure 4—figure supplement 2 (from Figure 2-4) so that they are there for examination, but do not emphasize them.

b) Sources of variability. First, we emphasize that in response probability and latency, the AIA response to a strong diacetyl stimulus is highly reproducible, and very low compared to any interneuron we or others have studied; it is only the weak stimuli that uncover its variable features. Second, we added more heatmaps to primary Figure 2 and supplementary figures associated with Figures 1, 2, 4 and 5, as requested by reviewer 2. Although heatmaps have their own shortcomings; the combination with traces in Figure 1 and Figure 1—figure supplement 2 should provide clarity. Third, we add additional information describing the analysis as the reviewer suggested; for example, response latency is not correlated across trials, but there’s a small decrease in response probability across trials; latency and probability are not correlated with GCaMP expression levels (Figure 1—figure supplement 3). Again, we do not emphasize response magnitudes, but Figure 1—figure supplement 2 shows that magnitudes are consistently smaller for spontaneous events than odor-evoked events at all thresholds, and Figure 4—figure supplement 2 shows that magnitudes are anti-correlated with GCaMP expression levels.

c) In all figures and supplementary figures, we downsampled for primary traces and heatmaps at random; there was no selection of traces except to represent the population at the level of percentage response and latency of response. We downsample heatmaps only for visibility, which is necessary especially when different experiments have different n values.

(d) Simultaneous measurements of multiple neurons. We have successfully obtained 20 events in which we simultaneously measure AIA, AWA, and ASK responses to diacetyl. These results are presented in Figure 5—figure supplement 2, and are consistent with our conclusions, but technical concerns are an issue here. The decreases in fluorescence in ASK are harder to threshold than the increases in AWA and AIA, because of spontaneous ASK activity and the slow offset kinetics of GCaMP reporters. To obtain orthogonal results supporting the AND-gate model, we have added electrophysiological experiments demonstrating a plausible basis of the observed AIA non-linearity and a correlation of the all-or-none plateau state with an all-or-none calcium signal (Figure 6). We believe that taking this entirely separate experimental approach is conceptually and technically stronger than further calcium imaging experiments.

2) Temporal filtering by AIAA second, more minor, conclusion drawn in the Discussion is that AIA integration serves to filter out transient or noisy odor stimuli. However, this idea is not explicitly tested in the manuscript. Odor dynamics can be challenging to control but can be done with proportional valves and measured by photo-ionization detector. Chrimson activation can be reliably controlled in time. If the authors wish to draw conclusions about temporal filtering or noise rejection in AIA I think they need to test this idea by explicitly varying the timecourse of odor or the level of background noise and measuring responses.

This was a side point in the Discussion and we have de-emphasized it.

Reviewer #2:1) Subsection “Calcium Imaging”, fourth paragraph: Both pulses were pooled for analyses.For both heatmaps in Figure 1C last panel (AWA:Chr) and Figure 1—figure supplement 1E, it appears that probability of AIA response is <50% (with varying latencies) given each row is a single trial (n= 569). Number of animals tested (Figure 1—figure supplement 1J) are 282 where 35% respond to both AWA:Chr pulses, 28% to 1st only, and 16% to 2nd only. What is getting pooled in Figure 1? Only traces which respond to both pulses?

Figure 1 includes AIA responses to AWA::Chrimson from all experiments with wild-type animals (n=569 stimulus pulses, 287 animals), regardless of whether the animal responded to the first, second, both, or neither stimulus pulse. In Figure 1—figure supplement 3, we analyzed responses from 282 of the 287 animals in which usable results were obtained during both trials. We clarify this information in the figure legends.

2) The mean trace for the AWA:Chr stimulus is similar to 115 nM diacetyl, but the trials seem more variable, including a number of non-responders. Are these most likely expression differences, or something about Chrimson dynamics? Can you be sure the low-level of 474 nm blue light used to image GCaMP prior to red light stimuluation does not have an effect on Chrimson? Did you try stimulating only the dendrite / sensory ending of AWA?

Yes. The AWA::Chrimson transgene has some variability that we ascribe to animal-to-animal transgene failure, which is not unusual in *C. elegans*. In experiments combining GCaMP with Chrimson in AWA, we found that 32/38 animals had GCaMP responses in 4/4 Chrimson-light trials, but 6/38 animals failed to respond to any of 4 Chrimson-light stimuli (Figure 1—figure supplement 1E). Looking across all AWA:Chrimson -> AIA results in the paper, they all still yield significant and consistent results if they are assumed to include transgene failure in ~15% of the animals. For example, in Figure 1—figure supplement 3H, one can assume that the 22% of animals that respond to neither stimulus include a respectable number of transgene failures, but that wouldn’t explain the 44% of animals that responded to only one of two light pulses. We now make that point in the Figure 1—figure supplement 3 legend. We couldn’t explain the issue by level of transgene expression, see Figure 1—figure supplement 1F. We did not attempt to stimulate dendrites. On the second point on simultaneous GCaMP imaging and Chrimson stimulation – yes, that was a challenge worked out during methods development. See response to main points above, and Figure 1—figure supplement 1D documenting methods.

3) Did you attempt to image AWA and AIA at the same time, or is this precluded by the need to image the AIA process?

We added a limited number of simultaneous recordings of AIA, AWA, and ASK to Figure 5—figure supplement 2. The results are consistent with our conclusions, but technical concerns are an issue here. The decreases in fluorescence in ASK are harder to threshold than the increases in AWA and AIA, both because of spontaneous ASK activity and because of the slow offset kinetics of GCaMP reporters.

4) unc-13 mutants as well another transgene of unc-18 seem to have no effect on AIA response latency at 1.15uM diacetyl. Though this is mentioned in the second paragraph of the subsection “Chemical synapses inhibit AIA”, the rationale is not clearly explained. Is the reliability of the response also dosage dependent? Only data related to latency are shown in Figure 3A and Figure 3—figure supplement 1A, heatmaps which can indicate reliability are missing.

We view the smaller effect of *unc-13* and *unc-18* mutations on AIA responses as the result of a ceiling effect. We note that all three chemical synapse mutants tested had slightly faster responses than wildtype, although only one reached statistical significance. We have added this point to the main text.

5) As in comment 2, only response time profiles are shown in Figure 4, indicating differences in latency, but heatmaps should be added as supplementary data to show trial-to-trial variability in responses.

At the reviewer’s request, we added heat maps for all experiments presented in Figure 4, to Figure 4—figure supplement 1.

6) At 11.5nM diacetyl, AWA is activated and ASK is inhibited, and this leads to moderate reliability of AIA response. At 1.15uM dia, AWA is activated with high fidelity, ASE is activated; ASK and AWC are inhibited and this leads to disinhibition of AIA and high reliability in its response. Do glutamate levels or different channel expression regulate the magnitude of disinhibition and in turn affect the reliability? ASK seems to be a key candidate tuning this reliability, as inhibition of ASK alone (Figure 4D) has a significant effect on AIA response reliability.

We don’t have insight into glutamate levels or the specific glutamate receptors involved in parsing the signals from the different sensory neurons.

7) Glutamate release from ASE (with AWC::Figure 4I) can inhibit AIA activation. However, 1.15uM dia activates ASE, yet increases AIA reliability. This is not well clarified in the manuscript, and a model figure and extended discussion would help in this regard.

ASE is complicated, so we’ve de-emphasized it in the figures while addressing its role further in the main text. Yes, ASE responds to diacetyl with depolarization, but this ASE response appears to be indirect, since it’s blocked in an *unc-18* synaptic mutant, unlike all other sensory neuron responses. The sensory neurons are highly interconnected, but we don’t know which one is important for activating ASE except that it’s not AWA. As the reviewer notes, the clear results we have are for ASK (diacetyl) and AWC (high diacetyl and isoamyl alcohol). We have reworked the model figures to emphasize this analysis, and think this should be helpful.

8) Were the unc-9 and unc-7 mutants tested individually to identify which of the innexins are required for AWA-AIA communication?

*unc-7* and *unc-9* single mutants are not defective in AWA to AIA signaling. We added these results to Figure 2D, and added cell-selective AWA/AIA *unc-9* rescue in the double mutant to Figure 2F.